# Diffusion Model as a Noise-Aware Latent Reward Model for Step-Level Preference Optimization

**Tao Zhang**[1,3]    **Cheng Da**[2]    **Kun Ding**[1]    **Huan Yang**[2]    **Kun Jin**[2]    **Yan Li**[2]

**Tingting Gao**[2]    **Di Zhang**[2]    **Shiming Xiang**[1,3]    **Chunhong Pan**[1]

[1]MAIS, Institute of Automation, Chinese Academy of Sciences

[2]Kuaishou Technology        [3]School of Artificial Intelligence, UCAS

## Abstract

Preference optimization for diffusion models aims to align them with human preferences for images. Previous methods typically use Vision-Language Models (VLMs) as pixel-level reward models to approximate human preferences. However, when used for step-level preference optimization, these models face challenges in handling noisy images of different timesteps and require complex transformations into pixel space. In this work, we show that pre-trained diffusion models are naturally suited for step-level reward modeling in the noisy latent space, as they are explicitly designed to process latent images at various noise levels. Accordingly, we propose the **Latent Reward Model (LRM)**, which repurposes components of the diffusion model to predict preferences of latent images at arbitrary timesteps. Building on LRM, we introduce **Latent Preference Optimization (LPO)**, a step-level preference optimization method conducted directly in the noisy latent space. Experimental results indicate that LPO significantly improves the model's alignment with general, aesthetic, and text-image alignment preferences, while achieving a 2.5-28$\times$ training speedup over existing preference optimization methods. Our code and models are available at `https://github.com/Kwai-Kolors/LPO`.

## 1   Introduction

Diffusion models [1] have achieved significant success in the domain of text-to-image generation. Inspired by advancements in preference optimization [2, 3] for Large Language Models (LLMs), several methods are proposed to align the diffusion model with human preferences. Diffusion-DPO [4] extends Direct Preference Optimization (DPO) [5] to diffusion models, leveraging human-annotated data for training without requiring a reward model. However, the reliance on offline sampling introduces a distribution discrepancy between the preference data and the model, resulting in reduced optimization effectiveness. In contrast, DDPO [6] and D3PO [7] employ online sampling and reward models for preference assessment. Furthermore, to address the issue of preference order inconsistency at different timesteps, Step-by-step Preference Optimization (SPO) [8] introduces the Step-aware Preference Model (SPM) to predict step-wise preferences. Consequently, SPO can align models with aesthetic preference through step-level online sampling and preference assignment.

Reward models are critical for the aforementioned methods and are mainly implemented by fine-tuning Vision-Language Models (VLMs) like CLIP [9] on preference datasets. We designate these models as Pixel-level Reward Models (PRMs) since they exclusively accept pixel-level image inputs. When applied to step-level preference optimization, PRMs encounter several common challenges. (1) **Complex Transformation**: At each timestep $t$, PRMs necessitate additional processes of diffusion denoising and VAE [10] decoding to transform noisy latent images $x_t$ into clean ones $\hat{x}_{0,t}$ and

---

✉Corresponding author: Kun Ding <kun.ding@ia.ac.cn>.

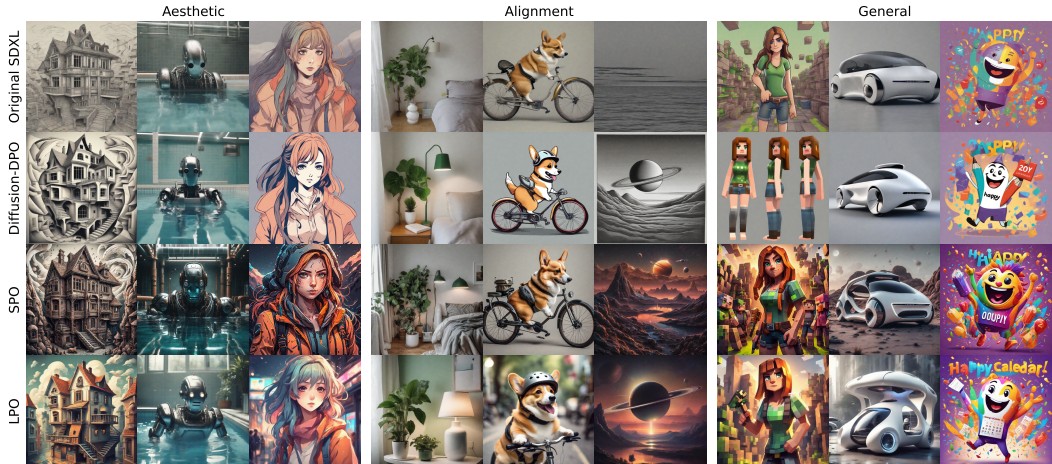

Figure 1: Qualitative comparison among different preference optimization methods based on SDXL [11]. LPO excels in both aesthetics and text-image alignment, resulting in improved overall quality. Larger versions with specific prompts are provided in Fig. 12 and Fig. 13.

pixel-level images $I_t$. This results in an overly lengthy inference process, as illustrated in Fig. 2 (a) and Fig. 4 (a). (2) **High-Noise Incompatibility**: At large timesteps characterized by high-intensity noise, the predicted images $I_t$ are significantly blurred (as illustrated in Fig. 9), leading to a severe distribution shift from the training data of VLMs, *i.e.*, clear images, thus making the predictions of PRMs at large timesteps unreliable. (3) **Timestep Insensitivity**: Since PRMs typically do not incorporate timesteps as input, it is challenging to understand the impact of different timesteps on image assessment. These issues hinder the effectiveness of PRMs for step-level reward modeling.

*Is there a model that can naturally capture human preferences directly in the latent space while being aware of timesteps and compatible with high-intensity noise?* **We find that the pre-trained diffusion model for text-to-image generation is an ideal choice because it exhibits several favorable characteristics**, as listed in Fig. 2 (b). (1) It possesses inherent text-image alignment capabilities, owing to the pre-training on large-scale text-image pairs. (2) It can directly process noisy latent images $x_t$ without requiring additional diffusion forwarding and VAE decoding. (3) It is high-noise-compatible since it can extract features from $x_t$ with various noise intensities, as is done during pre-training. (4) It naturally exhibits a strong sensitivity to the denoising timestep, enabling it to effectively grasp the model's attention at different timesteps. These pre-trained abilities make the diffusion model particularly suitable for step-level reward modeling in the noisy latent space.

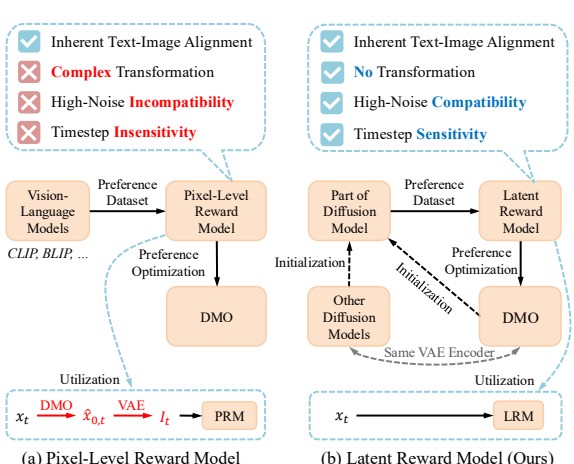

Figure 2: The comparison between the pixel-level reward model (a) and the latent reward model (b). DMO denotes the diffusion model to be optimized.

Based on the above insights, we propose using the diffusion model as a noise-aware **Latent Reward Model (LRM)**. Given noisy latent images $x_t$, LRM utilizes visual features from U-Net [12] or DiT [13] and textual features from the text encoder to predict step-level preference labels. A Visual Feature Enhancement (VFE) module is employed to enhance LRM's focus on the text-image alignment. To address the inconsistent preference issue in LRM's training data, we propose the Multi-Preference Consistent Filtering (MPCF) strategy to ensure that winning images consistently outperform losing ones across multiple dimensions. Finally, we employ LRM for step-level preference optimization, leading to a simple yet effective method termed **Latent Preference Optimization (LPO)**, where all steps are conducted within the noisy latent space of diffusion models. Extensive experiments on SD1.5 [14] and SDXL [11] demonstrate that LPO substantially improves the quality of generated

images and consistently outperforms existing DPO and SPO methods across the general, aesthetic, and alignment preferences, as indicated in Fig. 1. Meanwhile, LPO exhibits remarkable training efficiency, achieving a speedup of 10-28$\times$ over Diffuison-DPO [4] and 2.5-3.5$\times$ over SPO [8]. Furthermore, we explore a step-wise variant of GRPO [15] based on LRM and apply LPO to the DiT-based SD3 [1] model, demonstrating the generalization ability of LRM and LPO.

The core contributions of this paper are summarized as follows: (1) A noise-aware Latent Reward Model is introduced, which repurposes the pre-trained diffusion model for step-level reward modeling in the noisy latent space. (2) A Multi-Preference Consistent Filtering strategy is proposed to refine the public preference dataset, enabling LRM to better align with human preferences. (3) A Latent Preference Optimization method based on LRM is introduced to perform step-level preference optimization directly within the noisy latent space of diffusion models. (4) Extensive experimental results demonstrate the effectiveness, efficiency, and generalization ability of the proposed methods.

## 2   Related Work

**Reward Models for Image Generation.** Evaluating text-to-image generative models is a challenging problem. Several methods leverage VLMs to assess the alignment of images with human preferences. PickScore [16], HPSv2 [17], and ImageReward [18] aim to predict general preference by fine-tuning CLIP [9] or BLIP [19] on preference datasets. MPS [20] is proposed to capture multiple preference dimensions. Based on PickScore, SPM [8] is introduced to predict step-level preference labels during the denoising process. Recently, some works [21, 22, 23] employ more powerful VLMs to learn human preferences. However, these reward models, limited to accepting pixel-level images, often face problems like distribution shift and cumbersome inference when used in step-level preference optimization. In contrast, our proposed LRM can effectively mitigate these issues. To the best of our knowledge, we are the first to employ diffusion models themselves for reward modeling.

**Preference Optimization for Diffusion Models.** Motivated by improvements in Reinforcement Learning from Human Feedback (RLHF) in LLMs [2, 3], several optimization approaches for diffusion models have been proposed. Differentiable reward fine-tuning methods [24, 25, 26, 27] directly adjust diffusion models to maximize the reward of generated images. However, they are susceptible to reward hacking issues and require gradient backpropagation through multiple denoising steps. DPOK [28] and DDPO [6] formulate the denoising process as a Markov decision process and employ Reinforcement Learning (RL) techniques for preference alignment, but exhibit inferior performance on open vocabulary sets. Many works [4, 7, 29, 30, 31, 32] apply DPO [5] in LLMs to diffusion models, yielding better performance than the aforementioned RL-based methods. To mitigate the issue of inconsistent preference order across different timesteps, SPO [8] proposes a step-level preference optimization method. In addition, some works [33, 34] also explore training-free preference optimization methods. These methods typically perform optimization in pixel space, necessitating complex transformations and encountering distribution shift at large timesteps. On the contrary, LPO optimizes diffusion models directly in the noisy latent space using the LRM as a powerful and cost-effective reward model, demonstrating significant effectiveness and efficiency.

## 3   Preliminaries

### 3.1   Latent Diffusion Models

The forward process of diffusion models gradually adds random Gaussian noise to clean latent images $x_0$ to obtain noisy latent images $x_t$ at timestep $t$, in the manner of a Markov chain:

$$q(x_t|x_{t-1}) = \mathcal{N}(\sqrt{\alpha_t}x_{t-1}, (1 - \alpha_t)\mathbf{I}), \quad \alpha_t = 1 - \beta_t, \tag{1}$$

where $\mathcal{N}(\mu, \Sigma)$ denotes the Gaussian distribution and $\beta_t$ is a pre-defined time-dependent variance schedule. $\mathbf{I}$ denotes the identity matrix. The backward process aims to denoise $x_t$, which can be formulated as follows according to DDIM [35]:

$$p_\theta(x_{t-1}|x_t) = \mathcal{N}(\mu_t, \sigma_t^2 \epsilon_t), \quad \epsilon_t \sim \mathcal{N}(\mathbf{0}, \mathbf{I}), \tag{2}$$

$$\mu_t = \sqrt{\bar{\alpha}_{t-1}}\hat{x}_{0,t} + \sqrt{1 - \bar{\alpha}_{t-1} - \sigma_t^2} \cdot \epsilon_{\theta,t}, \quad \hat{x}_{0,t} = (\frac{x_t - \sqrt{1 - \bar{\alpha}_t}\epsilon_{\theta,t}}{\sqrt{\bar{\alpha}_t}}), \tag{3}$$

$$\sigma_t = \eta \sqrt{(1 - \bar{\alpha}_{t-1})/(1 - \bar{\alpha}_t)} \sqrt{1 - \alpha_t}, \quad \bar{\alpha}_t = \prod_{i=1}^{t} \alpha_t, \tag{4}$$

where $\eta \in [0, 1]$ is a hyperparameter to adjust the standard deviation $\sigma_t$. $\epsilon_{\theta,t}$ and $\hat{x}_{0,t}$ denote the noise and clean latent images predicted by diffusion models with parameters $\theta$ at timestep $t$.

### 3.2 Preference Optimization for Diffusion Models

Given winning latent images $x_0^w$, losing ones $x_0^l$, and condition $c$, Diffusion-DPO [4] propagates the preference order of $(x_0^w, x_0^l)$ to all denoising steps, thereby freely generating intermediate preference pairs $(x_t^w, x_t^l)$. It encourage models $p_\theta$ to generate $x_t^w$ rather than $x_t^l$ by minimizing

$$L_{DPO} = -\mathbb{E}_{\substack{x_{t,t+1}^w \sim p_\theta(x_{t,t+1}^w | x_0^w, c), \\ x_{t,t+1}^l \sim p_\theta(x_{t,t+1}^l | x_0^l, c)}} \left[ \log \sigma \left( \beta \log \frac{p_\theta(x_t^w | x_{t+1}^w, c)}{p_{ref}(x_t^w | x_{t+1}^w, c)} - \beta \log \frac{p_\theta(x_t^l | x_{t+1}^l, c)}{p_{ref}(x_t^l | x_{t+1}^l, c)} \right) \right], \tag{5}$$

where $\beta$ is a regularization hyperparameter and $p_{ref}$ denotes the reference model, fixed to the initial value of $p_\theta$. However, the preference order along the denoising process is not always identical. Therefore, SPO [8] proposes SPM to predict preferences for intermediate steps, and sample the win-lose pairs $(x_t^w, x_t^l)$ from the same $x_{t+1}$ to make the two paths comparable. Accordingly, the optimization objective of SPO is reformulated as minimizing

$$L_{SPO} = -\mathbb{E}_{x_t^w, x_t^l \sim p_\theta(x_t | x_{t+1}, c)} \left[ \log \sigma \left( \beta \log \frac{p_\theta(x_t^w | x_{t+1}, c)}{p_{ref}(x_t^w | x_{t+1}, c)} - \beta \log \frac{p_\theta(x_t^l | x_{t+1}, c)}{p_{ref}(x_t^l | x_{t+1}, c)} \right) \right]. \tag{6}$$

As a Pixel-level Reward Model (PRM), SPM faces some common issues discussed in Sec. 1. First, as illustrated in Fig. 4 (a) and (c), it requires additional procedures including $\hat{x}_{0,t}$ prediction and VAE decoding. Second, it struggles to process highly blurred $I_t$ when $t$ is large (Fig. 9). Third, although the AdaLN block [13] is introduced in SPM to improve its sensitivity to timesteps, insufficient pre-training makes it challenging to understand the focus of different timesteps on image generation. These issues jointly diminish the effectiveness of SPM in step-level preference optimization.

## 4 Method

To address the limitations of PRMs, we propose the Latent Reward Model (LRM) in Sec. 4.1, leveraging diffusion models for reward modeling. To facilitate its training, we propose a Multi-Preference Consistent Filtering (MPCF) strategy in Sec. 4.2. Finally, the Latent Preference Optimization (LPO) is introduced in Sec. 4.3 to optimize diffusion models directly in the noisy latent space.

### 4.1 Latent Reward Model

**Architecture of LRM.** LRM leverages features of the U-Net and text encoder in the diffusion model for preference prediction, as depicted in Fig. 3. Specifically, the textual features $f_p \in \mathbb{R}^{n_l \times n_p}$ are extracted from the prompt $p$ by the text encoder, where $n_l$ and $n_p$ denote the number of tokens and the dimension of textual features, respectively. Following CLIP [9], we use the last token $f_{eos} \in \mathbb{R}^{1 \times n_p}$ to represent the entire prompt and incorporate a text projection layer to obtain the final textual features $T \in \mathbb{R}^{1 \times n_d}$, where $n_d$ represents the final dimension of textual and visual features. Each noisy latent image $x_t$ is passed through the U-Net to interact with textual features $f_p$. The visual features of U-Net are average pooled along the spatial dimension, resulting in the multiscale

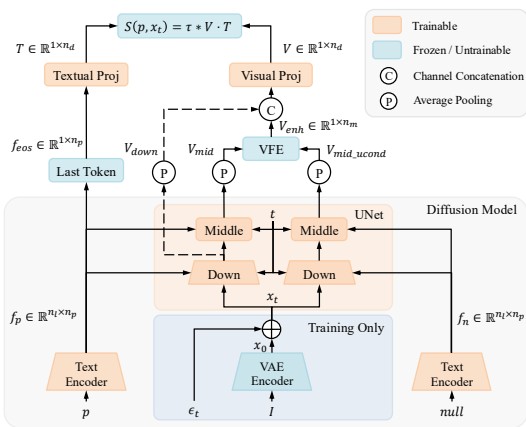

Figure 3: The architecture of LRM. The VAE encoder is only used during training.

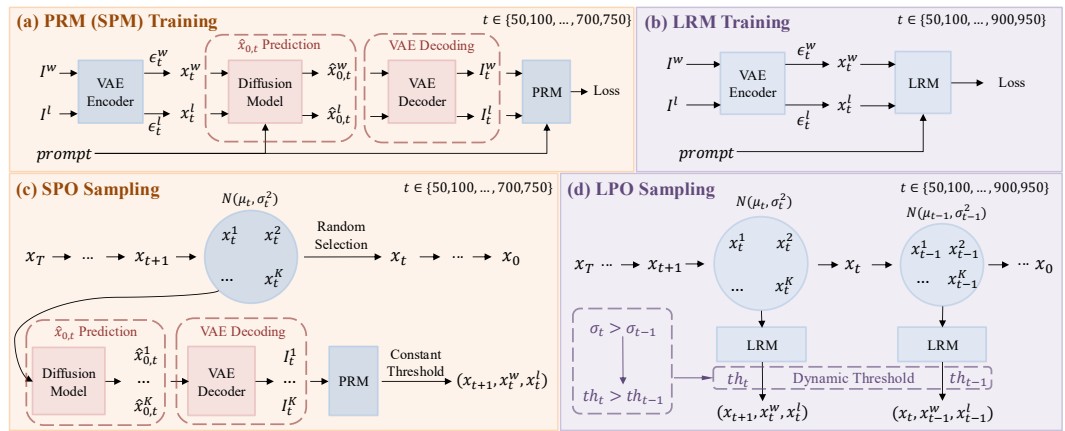

Figure 4: The training pipelines of PRM (a) and LRM (b), and sampling pipelines of SPO (c) and LPO (d). Compared to PRM, LRM can directly process latent images $x_t$ at various noise levels without requiring $\hat{x}_{0,t}$ prediction and VAE decoding. Therefore, LPO can perform sampling entirely within the noisy latent space throughout the full denoising process, covering $t \in [0, 950]$.

down-block features $V_{down}$ and middle-block features $V_{mid}$ as follows

$$V_{down}, V_{mid} = \text{AvgPool}(\text{U-Net}(x_t, f_p)), \quad V_{down} = \{V_{d_i}, i = 1, \ldots, L\}, \tag{7}$$

where $L$ is the number of down blocks and $V_{d_i}$ represents features of the $i$-th down blocks. However, it is observed that these features lack sufficient correlations with the textual features. Inspired by the Classifier-Free Guidance [36], we propose the **Visual Feature Enhancement (VFE)** module to enhance correlations between visual and textual features. It first extracts middle-block features $V_{mid\_ucond}$ without textual information, following a process similar to extracting $V_{mid}$, but with a null prompt. The enhanced visual features $V_{enh} \in \mathbb{R}^{1 \times n_m}$ are then computed as follows:

$$V_{enh} = V_{mid} + (gs - 1) * (V_{mid} - V_{mid\_ucond}), \tag{8}$$

where $gs \geq 1$ is a hyperparameter and $n_m$ is the dimension of $V_{mid}$. A larger $gs$ value injects more text-related features into $V_{mid}$, thus enhancing its focus on text-image alignment. When $gs = 1$, the VFE module is disabled. Next, $V_{enh}$ is concatenated with down-block features $V_{down}$ along the channel dimension and projected into the final visual features $V \in \mathbb{R}^{1 \times n_d}$ via a visual projection layer. Finally, the preference score of $x_t$ and $p$ is the dot product between textual and visual features:

$$S(p, x_t) = \tau * (l_2(V) \cdot l_2(T)), \tag{9}$$

where $\tau$ is a temperature coefficient following CLIP [9] and $l_2$ denotes the L2 Norm.

**Training Loss.** LRM is trained on the public preference dataset Pick-a-Pic v1 [16], denoted as $D$, which consists of triples, each containing a preferred image $I^w$, a less preferred image $I^l$, and the corresponding prompt $p$. As depicted in Fig. 4 (b), given uniformly sampled timestep $t \sim \mathcal{U}(0, T)$ where $T$ denotes the total denoising steps, the pixel-level image $I$ is encoded into the latent image $x_0$ using the VAE encoder. Noise $\epsilon_t \sim \mathcal{N}(\mathbf{0}, \mathbf{I})$ is then added to simulate $x_t$ in the backward denoising process. Following the Bradley-Terry (BT) model [37], the training loss of LRM is formulated as:

$$L_{LRM} = -\mathbb{E}_{t \sim \mathcal{U}(0,T), (I^w, I^l, p) \in D} \log \frac{exp(S(p, x_t^w))}{exp(S(p, x_t^w)) + exp(S(p, x_t^l))}, \tag{10}$$

$$x_t^* = \sqrt{\bar{\alpha}_t} * \text{VAE}_{\text{encoder}}(I^*) + \sqrt{1 - \bar{\alpha}_t} * \epsilon_t^*, \quad * \in \{w, l\}. \tag{11}$$

During training, the parameters of VAE are frozen to ensure that the latent space of the LRM remains stable, while the text encoder and U-Net are trainable.

## 4.2 Multi-Preference Consistent Filtering

**Issue of Inconsistent Preferences.** The training loss of LRM in Eqn. (10) involves the assumption that if $x_0^w$ is preferred over $x_0^l$, then after adding noise of equal intensity, the preference order remains unchanged, meaning $x_t^w$ continues to be preferred over $x_t^l$. However, this assumption breaks down

when the winning image excels in one aspect but is inferior in other aspects. For example, if the winning image $x_0^w$ has better details but exhibits weaker text-image alignment compared to the losing image $x_0^l$, the advantage in detail may be diminished after introducing significant noise, making the noisy losing image $x_t^l$ potentially more preferred than $x_t^w$. Such situations frequently occur in the Pick-a-Pic v1 [16] dataset. To demonstrate this, we employ the Aesthetic Score $S_A$ [38] to evaluate the aesthetic quality while using CLIP Score

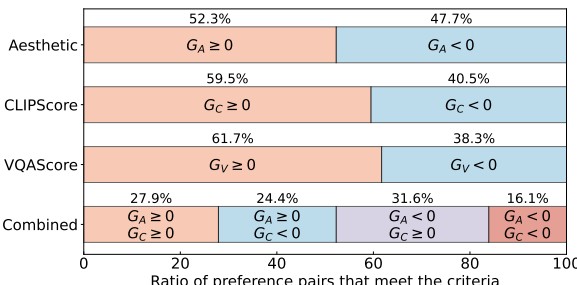

Figure 5: The distribution of preference consistency between human labels and reward scores in Pick-a-Pic.

$S_C$ [9] and VQAScore $S_V$ [39] to assess the text-image alignment. Let $G_* = S_*(I^w) - S_*(I^l), * \in \{A, C, V\}$ denote the preference score gaps between winning and losing images. As illustrated in Fig. 5, nearly half of the winning images have Aesthetic Scores lower than those of losing images. Similarly, around 40% of the winning images exhibit lower CLIP Scores and VQAScores. When considering both Aesthetic and CLIP scores, over 70% of the winning images score lower than the losing images in at least one aspect.

**Multi-Preference Consistent Filtering.**
We hypothesize that if the winning image outperforms the losing image across various aspects, the preference order is more likely to remain consistent after introducing similar noise. Based on this hypothesis, we propose the Multi-Preference

Table 1: Data filter strategies for Pick-a-Pic.

| Part | Strategy | Filter Rule | Num |
|------|----------|-------------|-----|
| Win-Lose | 1 | $G_A \geq 0, G_C \geq 0, G_V \geq 0$ | 101,573 |
| | 2 | $G_A \geq -0.5, G_C \geq 0, G_V \geq 0$ | 168,539 |
| | 3 | $G_A \geq -1, G_C \geq 0, G_V \geq 0$ | 201,885 |
| Tie | - | $|G_A| \leq 0.2, |G_C| \leq 0.03, |G_V| \leq 0.07$ | 8,537 |

Consistent Filtering (MPCF) strategy, which aims to approximate the condition of the above hypothesis by filtering data using various preference score gaps. Pick-a-Pic v1 contains 511,840 win-lose pairs and 71,907 tie pairs. Three strategies are explored for win-lose pairs, as detailed in Tab. 1. The first strategy is the most strict, but it is observed that it causes the LRM to overfit to the aesthetic aspect while neglecting the text-image alignment. In contrast, the third strategy is the most lenient regarding aesthetics, resulting in the LRM being unable to perceive the aesthetics of the images. It is found that the second strategy effectively balances both aspects, leading us to choose it ultimately. Further investigations and experiments are detailed in Sec. 5.3.

## 4.3 Latent Preference Optimization

To utilize LRM for step-level preference optimization, we propose the Latent Preference Optimization (LPO) method. We also explore the application of LRM on GRPO [15] in Section B.

**Sampling and Training.** As illustrated in Fig. 4 (d), at each timestep $t$, LPO samples a set of $x_t^i, i = 1, ..., K$ from the same $x_{t+1}$. These noisy latent images are then directly fed into LRM to predict preference scores $S_t^i, i = 1, ..., K$. The highest and lowest scores are normalized by the SoftMax function. If their score gap exceeds a predefined threshold $th_t$, the $x_t$ with the highest score is selected as $x_t^w$, while the one with the lowest score is designated as $x_t^l$, forming a qualified training sample $(x_{t+1}, x_t^w, x_t^l)$. Finally, these samples are used to optimize the diffusion model using the loss in Eqn. (6). As a result, **the entire process of LPO is conducted within the noisy latent space, eliminating the need for $\hat{x}_{0,t}$ prediction and VAE decoding**, which significantly enhances the training effectiveness and efficiency of LPO.

**Optimization Timesteps.** Due to the inaccuracy of SPM at high-noise levels, SPO samples training data only at low to medium noise levels, specifically when $t \in [0, 750]$, as shown in Fig. 4 (c). In contrast, owing to LRM's noise-aware ability, **LPO can perform sampling and training throughout the entire denoising process**, covering $t \in [0, 950]$. Experiments in Sec. 5.3 and Section B demonstrate that optimization at high noise levels plays a crucial role in preference optimization.

**Dynamic Threshold.** The sampling threshold $th$ is a crucial hyperparameter in LPO. A higher threshold tends to sample win-lose pairs with more pronounced differences, improving the quality of the training samples but reducing their overall quantity. Conversely, a lower threshold increases the number of training samples but may introduce a certain amount of noisy samples. Given that the

standard deviation $\sigma_t$ in Eqn. (2) decreases with $t$, using a constant threshold across all timesteps can hinder the training effectiveness. To handle this, we implement a dynamic threshold strategy, which sets lower thresholds for smaller timesteps, as follows:

$$th_t = \frac{\sigma_t - \sigma_{min}}{\sigma_{max} - \sigma_{min}} * (th_{max} - th_{min}) + th_{min}, \tag{12}$$

where $\sigma_{max}$ and $\sigma_{min}$ are the maximum and minimum values of $\sigma_t$. The hyperparameters $th_{max}$ and $th_{min}$ are used to adjust the range of thresholds.

**Selection of LRM in LPO.** To avoid confusion, we refer to the diffusion model optimized in LPO as DMO. Since LPO is performed within the latent space, which is determined by the VAE encoder, the VAE of LRM should be identical to that of DMO. Therefore, the specific architecture of LRM can be initialized from DMO or any other diffusion models that share the same VAE encoder with DMO, as depicted in Fig. 2 (b). The former is termed *homogeneous optimization* because LRM and DMO share the same architecture, whereas the latter is called *heterogeneous optimization*.

## 5 Experiments

### 5.1 Experimental Setup

The experiments are mainly conducted on SD1.5 [14] and SDXL [11] without refiner. The LRM is first trained on Pick-a-Pic and then used to fine-tune diffusion models through LPO. If not specified, we employ *homogeneous optimization*. Ablation experiments are conducted on SD1.5.

**LRM and LPO Training.** We denote the LRM based on SD1.5 and SDXL as LRM-1.5 and LRM-XL, respectively. They are trained on the filtered Pick-a-Pic v1 [16] as clarified in Sec. 4.2. The $gs$ in the VFE module is set to 7.5. The same 4k prompts in SPO are used for the LPO training, randomly sampled from the training set of Pick-a-Pic v1. The DDIM scheduler [35] with 20 inference steps is employed. We use all steps for sampling and training, *i.e.* $t \in [0, 50, ..., 900, 950]$. The dynamic threshold range $[th_{min}, th_{max}]$ is set to $[0.35, 0.5]$ for SD1.5 and $[0.45, 0.6]$ for SDXL. The $\beta$ in Eqn. (6) is set to 500 and the $K$ in the sampling process is set to 4. More details are in Section A.

**Baseline Methods.** We compare LPO with DDPO [6], D3PO [7], Diffusion-DPO [4], MaPO [29], SPO [8], and SePPO [30]. These methods are trained on Pick-a-Pic, which ensures a relatively fair comparison. We also include InterComp [31] for reference, which uses a higher-quality internal dataset. More details are provided in Section A.

**Evaluation Protocol.** We evaluate various diffusion models across three dimensions: general preference, aesthetic preference, and text-image alignment. The PickScore [16], HPSv2 [17], HPSv2.1 [17], and ImageReward [18] are utilized to assess the general preference. The aesthetic preference is evaluated using the Aesthetic Score [38]. Following [8], both general and aesthetic preferences are assessed on the validation unique split of Pick-a-Pic v1, which has 500 different prompts. For text-image alignment, we employ the GenEval [40] and T2I-CompBench++ [41] metrics. For detailed evaluation practice, please refer to Section A. Additionally, we propose two metrics to assess LRM's correlations with aesthetics and text-image alignment. Specifically, we calculate the score gaps $G_*, * \in \{A, C, L\}$ between winning and losing images, where $A$, $C$, $L$ represent Aesthetic, CLIP, and LRM. For LRM, the score is taken at $t = 0$. Then the Pearson Correlation Coefficient [42] between $G_L$ and $G_A$ is referred to as *Aes-Corr* while that between $G_L$ and $G_C$ is termed *CLIP-Corr*. They are evaluated on the validation unique and test unique splits of Pick-a-Pic v1.

### 5.2 Main Results

**Quantitative Comparison.** As indicated in Tab. 2 and Tab. 3, LPO outperforms other methods across various dimensions by a large margin on SD1.5, especially in general preference and text-image alignment. Based on SDXL, LPO even slightly surpasses InterComp, although the latter utilizes a superior dataset comprising images from SD3[1] and FLUX [43]. The user study results in Section B further confirm the superiority of LPO. We also validate the effectiveness of *heterogeneous optimization* in Tab. 4. Remarkably, fine-tuning SD2.1 using LRM-1.5 yields significant improvements across various aspects, demonstrating that an inferior diffusion model can effectively fine-tune an advanced model as long as they share the same VAE encoder. In contrast, applying LRM-1.5 for the LPO of SDXL is ineffective due to a distribution mismatch between the latent spaces of their VAE encoders.

Table 2: General and aesthetic preference scores on Pick-a-Pic validation unique set, along with GenEval [40] scores. * denotes the metrics are copied from [8]. Others are evaluated using the official model. If not specified, we use 20 inference steps. PaP denotes the Pick-a-Pic [16] dataset. The complete results on GenEval are provided in Tab. 25 and Tab. 26.

| Model | Method | Dataset | PickScore | ImageReward | HPSv2 | HPSv2.1 | Aesthetic | GenEval (20 Step) | GenEval (50 Step) |
|---|---|---|---|---|---|---|---|---|---|
| SD1.5 [14] | Original | - | 20.56 | 0.0076 | 26.46 | 24.05 | 5.468 | 42.56 | 42.53 |
| | DDPO* [6] | PaP v2 | 21.06 | 0.0817 | - | 24.91 | 5.591 | - | - |
| | D3PO* [7] | PaP v1 | 20.76 | -0.1235 | - | 23.97 | 5.527 | - | - |
| | Diff.-DPO [4] | PaP v2 | 20.99 | 0.3020 | 27.03 | 25.54 | 5.595 | 43.79 | 45.13 |
| | SPO* [8] | PaP v1 | 21.43 | 0.1712 | - | 26.45 | 5.887 | - | - |
| | SPO [8] | PaP v1 | 21.22 | 0.1678 | 26.73 | 25.83 | 5.927 | 40.46 | 41.53 |
| | SePPO [30] | PaP v2 | 21.25 | 0.5077 | 27.56 | 27.34 | 5.766 | 44.97 | 44.46 |
| | LPO (ours) | PaP v1 | **21.69** | **0.6588** | **27.64** | **27.86** | **5.945** | **48.39** | **48.77** |
| SDXL [11] | Original | - | 21.65 | 0.4780 | 27.06 | 26.05 | 5.920 | 49.40 | 52.29 |
| | Diff.-DPO [4] | PaP v2 | 22.22 | 0.8527 | 28.10 | 28.47 | 5.939 | 57.78 | 58.91 |
| | MaPO [29] | PaP v2 | 21.89 | 0.7660 | 27.61 | 27.44 | 6.095 | 51.59 | 52.80 |
| | SPO* [8] | PaP v1 | 23.06 | 1.0803 | - | 31.80 | 6.364 | - | 55.20 |
| | SPO [8] | PaP v1 | 22.70 | 0.9951 | 28.42 | 31.15 | 6.343 | 50.52 | 52.75 |
| | InterComp [31] | Internal | 22.63 | **1.2728** | **29.08** | 31.52 | 6.016 | 59.24 | 59.65 |
| | LPO (ours) | PaP v1 | **22.86** | 1.2166 | 28.96 | **31.89** | **6.360** | **59.27** | **59.85** |

Table 3: Quantitative results on T2I-CompBench++ [41] with 20 inference steps.

| Model | Method | Color | Shape | Texture | 2D-Spatial | 3D-Spatial | Numeracy | Non-Spatial | Complex |
|---|---|---|---|---|---|---|---|---|---|
| SD1.5 [14] | Original [14] | 0.3783 | 0.3616 | 0.4172 | 0.1230 | 0.2967 | 0.4485 | 0.3104 | 0.2999 |
| | Diff.-DPO [4] | 0.4090 | 0.3664 | 0.4253 | 0.1336 | 0.3124 | 0.4543 | **0.3115** | 0.3042 |
| | SPO [8] | 0.4112 | 0.4019 | 0.4044 | 0.1301 | 0.2909 | 0.4372 | 0.3008 | 0.2988 |
| | SePPO [30] | 0.4265 | 0.3747 | 0.4170 | 0.1504 | 0.3285 | 0.4568 | 0.3109 | 0.3076 |
| | LPO (ours) | **0.5042** | **0.4522** | **0.5259** | **0.1928** | **0.3562** | **0.4845** | 0.3110 | **0.3308** |
| SDXL [11] | Original [11] | 0.5833 | 0.4782 | 0.5211 | 0.1936 | 0.3319 | 0.4874 | 0.3137 | 0.3327 |
| | Diff.-DPO [4] | 0.6941 | 0.5311 | 0.6127 | 0.2153 | 0.3686 | 0.5304 | 0.3178 | 0.3525 |
| | MaPO [29] | 0.6090 | 0.5043 | 0.5485 | 0.1964 | 0.3473 | 0.5015 | 0.3154 | 0.3229 |
| | SPO [8] | 0.6410 | 0.4999 | 0.5551 | 0.2096 | 0.3629 | 0.4931 | 0.3098 | 0.3467 |
| | InterComp [31] | 0.7218 | 0.5335 | 0.6290 | 0.2406 | 0.3929 | 0.5395 | **0.3212** | 0.3659 |
| | LPO (ours) | **0.7351** | **0.5463** | **0.6606** | **0.2414** | **0.4075** | **0.5493** | 0.3152 | **0.3801** |

**Qualitative Comparison.** The qualitative comparisons of various methods are illustrated in Fig. 1 and Fig. 11-Fig. 15. Images generated by Diffusion-DPO exhibit deficiencies in color and detail, whereas those from SPO demonstrate reduced semantic relevance and excessive details in some images, resulting in cluttered visuals. In contrast, the images produced by LPO achieve a strong balance between text-image alignment and aesthetic quality, delivering a higher overall image quality.

**Training Efficiency Comparison.** LPO enables significantly faster training. As shown in Tab. 5, by performing reward modeling and preference optimization directly in the noisy latent space, LRM bypasses both $\hat{x}_{0,t}$ prediction and VAE decoding during sampling. When $K = 4$, its sampling time is only 1/6 that of SPM in SPO, yielding substantial time savings—and the same efficiency gain applies to LRM training. Consequently, as illustrated in Tab. 6, LPO requires only 23 A100 hours for SD1.5—just 1/10 the training time of Diffusion-DPO and 1/3.5 that of SPO. On SDXL, LPO's training time is reduced to 1/28 and 1/2.5 of that for Diffusion-DPO and SPO, respectively.

**Further Explorations.** In Section B, we present additional exploration experiments on LRM and LPO. First, we validate the effectiveness of LRM on a step-wise variant of GRPO [15]. We then assess the applicability of LPO on DiT-Based SD3 [1] and evaluate its generalization on other datasets. Additionally, we report the performance of LRM when used as a pixel-wise reward model.

## 5.3 Ablation Studies

**MPCF.** As shown in Tab. 7, MPCF plays a vital role in LRM training. The first filtering strategy enforces that winning images must outperform losing images across all aspects. However, since the diffusion model lacks explicit text-image alignment pre-training like CLIP, it is prone to overfitting to visual features, as indicated by a higher Aes-Corr. This overfitting results in reduced attention to alignment, as reflected by lower CLIP-Corr and GenEval scores. The second and third strategies relax the aesthetic constraints to varying degrees. While the third, most lenient strategy can cause LRM to focus solely on alignment and neglect image quality, as evidenced by a negative Aes-Corr value, the second strategy achieves a better balance, yielding the highest general preference scores.

Table 4: Heterogeneous optimization based on LRM-SD1.5.

| Model | Method | Aesthetic | GenEval | P-S | I-R | HPSv2 | HPSv2.1 |
|---|---|---|---|---|---|---|---|
| SD2.1 [14] (Same VAE) | Original | 5.673 | 48.59 | 20.92 | 0.3063 | 27.05 | 25.49 |
| | LPO (ours) | **5.969** | **56.01** | **21.76** | **0.7978** | **28.05** | **28.61** |
| SDXL [11] (Diff. VAE) | Original | 5.920 | **49.40** | **21.65** | **0.4780** | 27.06 | 26.05 |
| | LPO (ours) | **5.953** | 40.85 | 20.82 | 0.3919 | **27.10** | **26.69** |

Table 6: Comparisons of training speed between different methods.

| Method | Reward Modeling | Preference Optimization | Total ↓ (A100 h) |
|---|---|---|---|
| SD1.5 | | | |
| Diff.-DPO | 0 | 240 | 240 |
| SPO | 32 | 48 | 80 |
| LPO (ours) | **15** | **8** | **23** |
| SDXL | | | |
| Diff.-DPO | 0 | 2,560 | 2,560 |
| SPO | 116 | 118 | 234 |
| LPO (ours) | **52** | **40** | **92** |

Table 5: Time of each sampling step on a single A100.

| Reward Model | Denoising Step $(x_{t+1} \to x_t)$ | $\hat{x}_{0,t}$ Prediction $(x_t \to \hat{x}_{0,t})$ | VAE Decoding $(\hat{x}_{0,t} \to I_t)$ | Reward Prediction | Total ↓ (K=4) |
|---|---|---|---|---|---|
| SPM (PRM) | 0.019s | $K \times 0.019$s | $K \times 0.031$s | $K \times 0.006$s | 0.243s |
| LRM (ours) | 0.019s | 0 | 0 | $K \times 0.005$s | **0.039s** |

Table 7: Ablation results on LRM's training data.

| Strategy | LRM | | LPO | | |
|---|---|---|---|---|---|
| | Aes-Corr | CLIP-Corr | Aesthetic | GenEval | PickScore |
| wo MPCF | 0.1342 | 0.2274 | 5.772 | 45.66 | 21.49 |
| 1 | **0.4860** | 0.1011 | **6.390** | 45.77 | _21.61_ |
| 2 | 0.1136 | 0.3588 | _5.945_ | 48.39 | **21.69** |
| 3 | -0.1152 | **0.4480** | 5.750 | **48.62** | 21.47 |

Table 8: Ablation results on the VFE module.

| VFE | $gs$ | LRM | | LPO | | |
|---|---|---|---|---|---|---|
| | | Aes-Corr | CLIP-Corr | Aesthetic | GenEval | PickScore |
| ✗ | 1.0 | **0.1712** | 0.3211 | **6.053** | 46.60 | 21.51 |
| ✓ | 3.0 | 0.1233 | 0.3441 | 5.923 | 47.35 | 21.53 |
| ✓ | 7.5 | 0.1136 | 0.3588 | _5.945_ | 48.39 | **21.69** |
| ✓ | 10.0 | 0.1063 | **0.3592** | 5.937 | _48.13_ | _21.56_ |

Notably, even without MPCF and using the same training data, LPO still outperforms SPO in both general and text-image alignment preferences, as detailed in Section C.

**Structure of LRM.** As illustrated in Tab. 8, the introduction of VFE ($gs > 1$) leads to lower Aes-Corr values but higher CLIP-Corr values, indicating an enhanced emphasis on text-image alignment. This results in improvements in both the GenEval and PickScore, with only a minor decline in Aesthetic Score. As $gs$ increases, the LRM's correlation with alignment steadily improves, while its correlation with aesthetics decreases. When $gs$ is set to 7.5, the model achieves the best overall performance.

**Optimization Timesteps.** Tab. 9 ablates different optimization timestep ranges, indicating that broader ranges lead to better performance. Notably, LRM can effectively predict preferences even at very large timesteps, *e.g.*, $[750, 950]$, achieving results comparable to those obtained across the entire denoising process, *i.e.* $[0, 950]$. This highlights the critical role of large timesteps in step-level preference optimization. In contrast, SPO performs poorly in large timesteps, as indicated in Tab. 21 in Section C. This comparison underscores the superiority of using diffusion models for step-level reward modeling at large timesteps, since LRM can directly process noisy latent images without suffering from distribution shift. A qualitative comparison of different ranges is provided in Fig. 10.

**Dynamic Sampling Threshold.** The standard deviation $\sigma_t$ of samples at smaller timesteps is relatively small according to the DDPM scheduling [44], making the constant threshold insufficient to accommodate all timesteps. As indicated in Tab. 10, the dynamic threshold strategy generally outperforms the constant threshold across different intervals, effectively alleviating this problem. We further explore other dynamic strategies in Tab. 24 in Section C.

# 6 Discussion and Conclusion

**Conclusion.** In this paper, we propose LRM, a method that utilizes diffusion models for step-level reward modeling directly in the noisy latent space, based on the insights that diffusion models inherently possess text-image alignment abilities and can effectively perceive noisy latent images across different timesteps. To facilitate LRM training, we introduce the MPCF strategy to mitigate the inconsistent preference issue in training data. Building on LRM, we further propose LPO, a step-level preference optimization that operates entirely within the noisy latent space. LPO not only achieves substantial training speedups but also delivers remarkable performance improvements across various evaluation metrics, highlighting the effectiveness of employing the diffusion model itself to guide its preference optimization. We hope our findings can open new avenues for research in preference optimization for diffusion models and contribute to advancing the field of visual generation.

**Limitations.** (1) The Pick-a-Pic dataset mainly contains images generated by SD1.5 and SDXL, which generally exhibit low image quality. Introducing higher-quality images is expected to enhance

Table 9: Ablation results on timestep ranges.

| Range of $t$ | P-S | I-R | HPSv2 | HPSv2.1 | Aesthetic | GenEval |
|---|---|---|---|---|---|---|
| [0, 200] | 20.46 | -0.0987 | 26.25 | 23.61 | 5.434 | 40.11 |
| [250, 450] | 20.76 | 0.1430 | 26.90 | 25.37 | 5.527 | 43.00 |
| [500, 700] | 20.95 | 0.1591 | 26.71 | 25.16 | 5.742 | 44.44 |
| [750, 950] | 21.54 | 0.6337 | 27.47 | 27.64 | 5.853 | 48.28 |
| [0, 450] | 20.63 | 0.0204 | 26.69 | 24.88 | 5.573 | 42.71 |
| [0, 700] | 21.02 | 0.3087 | 27.10 | 26.25 | 5.765 | 44.93 |
| [0, 950] | 21.69 | 0.6588 | 27.64 | 27.86 | 5.945 | 48.39 |

Table 10: Ablation results on threshold strategies.

| Threshold | P-S | I-R | HPSv2 | HPSv2.1 | Aesthetic | GenEval |
|---|---|---|---|---|---|---|
| 0.3 | 21.22 | 0.5112 | 27.30 | 27.12 | 5.853 | 46.75 |
| 0.4 | 21.32 | 0.4789 | 27.08 | 26.37 | 5.832 | 48.32 |
| 0.5 | 21.57 | 0.6088 | 27.54 | 27.42 | 5.900 | 48.39 |
| 0.6 | 21.35 | 0.5510 | 27.25 | 26.73 | 5.877 | 47.97 |
| [0.3, 0.45] | 21.58 | 0.6405 | 27.55 | 27.33 | 5.916 | 49.43 |
| [0.35, 0.5] | 21.69 | 0.6588 | 27.64 | 27.86 | 5.945 | 48.39 |
| [0.4, 0.55] | 21.48 | 0.4791 | 27.30 | 27.13 | 5.882 | 48.77 |

the generalization of the LRM. (2) MPCF relies on three reward models to approximate human preferences for automatic data filtering. However, the filtered data may still inherit biases or limitations shared by these reward models. (3) Since LPO is performed within the latent space, which is determined by the VAE encoder, the VAE of LRM should be identical to that of DMO.

**Future Work.** (1) As a step-level reward model, the LRM can be easily applied to reward fine-tuning methods [45, 24, 26], avoiding lengthy inference chain backpropagation and significantly accelerating the training speed. (2) The LRM can also extend the best-of-N approach to a step-level version, enabling exploration and selection at each step of image generation, thereby achieving inference-time optimization similar to GPT-o1 [46].

## Acknowledgments and Disclosure of Funding

This research was supported by the National Natural Science Foundation of China under Grants 62433003 and 62306310.

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

## A  Experimantal Details

**LRM Training.** We train LRM-1.5 for 4,000 steps with 500 warmup steps and LRM-XL for 8,000 steps with 1000 warmup steps, both using the learning rate 1e-5. The batch size is 128 for LRM-1.5 and 32 for LRM-XL, respectively. Following CLIP [9], the initial value of $\tau$ in Eqn. (9) is set to $e^{2.6592}$. The training resolution is 512 for SD1.5 and 1024 for SDXL. For SDXL, which includes two text encoders, we utilize only the OpenCLIP ViT-bigG [47] as the text encoder of LRM-XL.

**LPO Training.** Both SD1.5 and SDXL are fine-tuned for 5 epochs. Following SPO [8], we employ LoRA [48] to fine-tune models. The LoRA rank is 4 for SD1.5 and 64 for SDXL. All experiments are conducted using 4 A100. Other hyperparameters are provided in Tab. 11. The experimental setting of SD2.1 is the same as SD1.5.

Table 11: Hyperparameters of LPO training.

|  | SD1.5 [14] | SDXL [11] |
|---|---|---|
| Learning Rate | 5e-5 | 1e-5 |
| LoRA Rank | 4 | 64 |
| $\beta$ | 500 | 500 |
| $K$ | 4 | 4 |
| Sampling Threshold Range | [0.35, 0.5] | [0.45, 0.6] |
| Sampling Batch Size | 5 | 4 |
| Training Resolution | 512×512 | 1024×1024 |
| Training Epoch | 5 | 5 |
| Training Batch Size | 10 | 4 |

**Baseline Methods.** Specifically, SPO [8] and LPO are trained on Pick-a-Pic v1 [16] while Diffusion-DPO [4], SePPO [30], and MaPO [29] are trained on Pick-a-Pic v2 [16], an extended version of Pick-a-Pic v1. Pick-a-Pic v1 consists of 583,737 preference pairs, whereas Pick-a-Pic v2 contains over 950,000 pairs. In the practice of SPO, DDPO [6] and D3PO [7] are reproduced on 4k prompts randomly sampled from Pick-a-Pic v1, which are the same as the training data used for SPO and LPO. Therefore, the comparison between these methods can be considered fair. Furthermore, for reference purposes, we also make a comparison with InterComp [31]. Instead of Pick-a-Pic, InterComp utilizes an internally constructed preference dataset of higher quality, including images generated by more advanced models such as SD3 [1] and FLUX [43]. In contrast, the data in Pick-a-Pic only consists of images from models like SDXL [11] and SD2.1 [14]. Despite the relatively low-quality training data, LPO still performs slightly better than InterComp, demonstrating the effectiveness of our method.

**Evaluation Practice.** For all baseline methods, we use the official models for evaluation without retraining. For the evaluation of text-image alignment, we utilize official codebases of GenEval [40] and T2I-CompBench++ [41]. Following SPO, we employ several public preference models to assess general and aesthetic preferences, including PickScore [16], ImageReward [18], HPSv2 [17], HPSv2.1 [17], and Aesthetic Predictor [38]. Since SPO does not provide the evaluation code, we implement our pipeline, adhering to SPO's setting: using the DDIM [35] scheduler with 20 inference steps and the same random seed to ensure reproducibility. We apply the same code for all models to ensure a fair comparison. Although there are some discrepancies with SPO's reported results, we believe the relative comparisons between different models are reliable and meaningful.

## B  Additional Exploration Experiments

**User Study.** We conduct human evaluation experiments for LPO, SPO, and Diffusion-DPO. The evaluation set consists of 200 prompts, with 100 randomly sampled from HPSv2 benchmark [17] and 100 randomly sampled from DPG-Bench [49]. For each prompt, the model generates four images. For each image, five expert evaluators score it across three dimensions: general preference, visual appeal, and prompt alignment. Subsequently, votes are cast to determine the winning relationships among the different models. The results are illustrated in Fig. 6. Similar to the results of automatic metrics in the main paper, LPO outperforms both SPO and Diffusion-DPO across three dimensions. Compared to T2I-CompBench and GenEval results, the advantage of LPO on text-image alignment in the human evaluation results is not particularly pronounced. We believe this is because the prompts

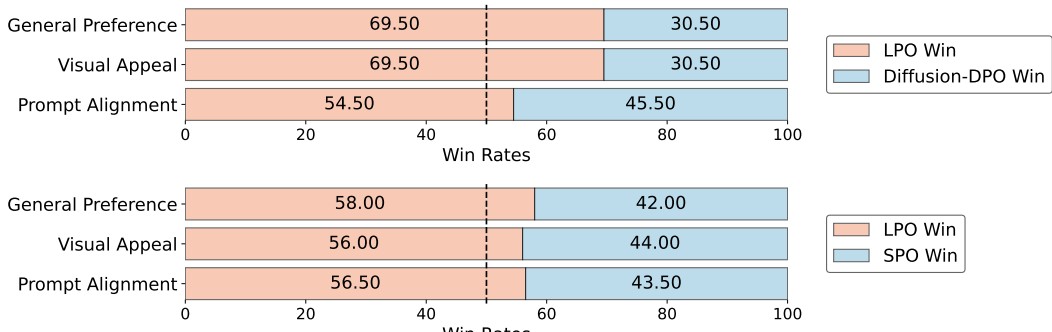

Figure 6: User study results on 200 prompts from HPSv2 benchmark [17] and DPG-Bench [49].

Table 12: The performance of different preference optimization methods based on LRM. The general and aesthetic scores are calculated on the Pick-a-Pic validation unique set. We use 20 inference steps.

| Model | Method | PickScore | ImageReward | HPSv2 | HPSv2.1 | Aesthetic | GenEval (20 Step) | Training Time ↓ (A100 hour) |
|---|---|---|---|---|---|---|---|---|
| SD1.5 [14] | Original | 20.56 | 0.0076 | 26.46 | 24.05 | 5.468 | 42.56 | - |
| | LPO | **21.69** | **0.6588** | **27.64** | 27.86 | 5.945 | **48.39** | 8 |
| | Step-Wise GRPO | 21.52 | 0.5861 | 27.47 | **28.35** | **6.087** | 45.88 | 64 |

Table 13: General and aesthetic preference scores on Pick-a-Pic validation unique set, along with GenEval [40] scores. We use 20 inference steps.

| Model | Method | Dataset | PickScore | ImageReward | HPSv2 | HPSv2.1 | Aesthetic |
|---|---|---|---|---|---|---|---|
| SD3-Medium [1] | Original | - | 21.74 | 0.9527 | 28.27 | 28.74 | 5.741 |
| | LPO | Pick-a-Pic v1 [16] | **22.29** | **1.1306** | **28.64** | **30.44** | **5.993** |

Table 14: Qualitative results on GenEval [40] with 20 inference steps based on SD3-Medium.

| Model | Method | Single Object | Two Object | Counting | Colors | Position | Color Attribution | Overall |
|---|---|---|---|---|---|---|---|---|
| SD3-Medium [1] | Original | 98.13 | 81.57 | 57.81 | 82.98 | 24.50 | **58.75** | 67.29 |
| | LPO | **100.00** | **86.11** | **60.63** | **84.04** | **26.25** | 54.75 | **68.63** |

used for human evaluation are not specifically designed to assess text-image alignment, and different models exhibit relatively good alignment.

**Exploration of GRPO Based on LRM.** Actually, LRM is not strictly bound to LPO, and it can also be applied to other preference optimization methods. Inspired by DeepSeekMath [15], we implement a step-wise GRPO algorithm for preference optimization in the image domain and use LRM as the reward model. Specifically, at each denoising step from $x_{t+1}$ to $x_t$, we sample a group of $x_t^i$, $i = 1, ..., K$ according to Eqn. (2). The LRM is employed to provide rewards $R_t = \{r_t^i, i = 1, ..., K\}$ for these noisy latent images. We then normalize these rewards to calculate corresponding advantages by $A_t^i = \frac{r_t^i - mean(R_t)}{std(R_t)}$. Finally, we optimize diffusion models by minimizing the following objective:

$$L_{GRPO} = -\mathbb{E}[\{x_t^i\}_{i=1}^K \sim p_\theta(x_t|x_{t+1}, c)]$$
$$\left\{ \min\left[ \frac{p_\theta(x_t^i|x_{t+1}, c)}{p_{\theta_{old}}(x_t^i|x_{t+1}, c)}, \text{clip}\left( \frac{p_\theta(x_t^i|x_{t+1}, c)}{p_{\theta_{old}}(x_t^i|x_{t+1}, c)}, 1 - \epsilon, 1 + \epsilon \right) \right] A_t^i - \beta \mathbb{D}_{KL}[p_\theta || p_{ref}] \right\} \quad (13)$$

We conduct preliminary experiments of step-wise GRPO on the SD1.5 model. We set $K$ to 4, $\beta$ to 0.1 and $\epsilon$ to 0.1. As shown in Tab. 12, based on our LRM-1.5, step-wise GRPO consistently improves the model's performance across various dimensions. However, in the current setting, step-wise GRPO performs slightly worse than LPO and requires $8\times$ the training time. We will further explore the application of LRM-based GRPO for image-domain preference optimization in future work.

**Exploration on DiT-Based Models with Flow Matching Methods (SD3).** In this paper, we mainly conduct experiments on U-Net-based models with the DDPM [44] scheduling method. Here we explore the effectiveness of LRM and LPO on DiT-based models with the Flow Matching method

Table 15: Performance of LPO on HPDv2. General and aesthetic preference scores on Pick-a-Pic validation unique set, along with GenEval [40] scores. If not specified, we use 20 inference steps.

| Model | Method | Dataset | PickScore | ImageReward | HPSv2 | HPSv2.1 | Aesthetic | GenEval (20 Step) | GenEval (50 Step) |
|-------|--------|---------|-----------|-------------|-------|---------|-----------|-------------------|-------------------|
| SD1.5 [14] | Original | - | 20.56 | 0.0076 | 26.46 | 24.05 | 5.468 | 42.56 | 42.53 |
| | LPO | HPDv2 [17] | **21.24** | **0.7248** | **28.13** | **28.93** | **5.917** | **47.29** | **48.27** |

Table 16: Performance of LPO on HPDv2. General and aesthetic preference scores on 400 randomly sampled HPDv2 test prompts with 20 inference steps.

| Model | Method | Dataset | PickScore | ImageReward | HPSv2 | HPSv2.1 | Aesthetic |
|-------|--------|---------|-----------|-------------|-------|---------|-----------|
| SD1.5 [14] | Original | - | 20.56 | 0.0076 | 26.46 | 24.05 | 5.468 |
| | LPO | HPDv2 [17] | **21.61** | **0.7282** | **28.81** | **29.25** | **6.019** |

Table 17: Performance of LPO on HPDv2. Quantitative results on T2I-CompBench++ [41] with 20 inference steps. LPO is trained on the HPDv2 [17] dataset.

| Model | Method | Color | Shape | Texture | 2D-Spatial | 3D-Spatial | Numeracy | Non-Spatial | Complex |
|-------|--------|-------|-------|---------|------------|------------|----------|-------------|---------|
| SD1.5 [14] | Original | 0.3783 | 0.3616 | 0.4172 | 0.1230 | 0.2967 | 0.4485 | **0.3104** | 0.2999 |
| | LPO | **0.5761** | **0.5067** | **0.6013** | **0.1671** | **0.3345** | **0.4671** | 0.3035 | **0.3332** |

[50]. Firstly, based on SD3-medium [1], we train the LRM on Pick-a-Pic v1 [16]. We extract visual features from the 18th layer of MMDiT [1] and utilize text features from all text encoders. The encoders of the SD-medium are frozen during training. Secondly, to provide the exploration space for LPO, we transform Flow Matching from ODE (Ordinary Differential Equation) to SDE (Stochastic Differential Equation). Specifically, we replace predicted noise $\epsilon_{\theta,t}$ in Eqn. (3) with predicted velocity $v_{\theta,t}$ using the relation $\epsilon_{\theta,t} = x_t + \alpha_t v_{\theta,t}$. Additionally, we substitute $\sqrt{\bar{\alpha}_t}$ and $\sqrt{1 - \bar{\alpha}_t}$ in Eqn. (3) and Eqn. (4) with $\alpha'_t$ and $\sigma'_t$ in the forward process of Flow Matching, $i.e.$, $x_t = \alpha'_t x_0 + \sigma'_t \epsilon$. Then the backward process can be reformulated as:

$$x_{t-1} = \alpha'_{t-1}(x_t - \sigma'_t v_{\theta,t}) + \sqrt{(\sigma'_{t-1})^2 - \sigma_t^2}(\alpha'_t v_{\theta,t} + x_t) + \sigma_t \epsilon_t, \quad \epsilon_t \sim \mathcal{N}(\mathbf{0}, \mathbf{I}), \quad (14)$$

$$\sigma_t = \eta \sqrt{\left(\frac{\sigma'_{t-1}}{\sigma'_t}\right)^2 \left(1 - \left(\frac{\alpha'_t}{\alpha'_{t-1}}\right)^2\right)}. \quad (15)$$

Finally, we employ LPO on SD3-medium. As shown in Tab. 13 and Tab. 14, even though the training data Pick-a-Pic only contains images from SDXL and SD2.1, LPO still demonstrates notable performance gains on various metrics. We believe that if there is higher-quality training data, such as that in InterComp, the performance gain will be more pronounced.

**Experimental Results on HPDv2.** To validate the generalization of LRM and LPO, we conduct experiments on the HPDv2 [17] dataset without any hyperparameter optimization. Specifically, we train LRM on the entire HPDv2 training set and then randomly sample 4,000 prompts from this set for LPO training. In addition to the previous evaluation metrics, we also randomly sample 400 prompts from the HPDv2 test set and calculate the general and aesthetic scores for these prompts. As shown in Tab. 15, Tab. 16, and Tab. 17, LPO significantly outperforms the original models across various evaluation dimensions.

**LRM as Pixel-Wise Reward Model.** The LRM can also serve as a pixel-wise reward model when combined with the corresponding VAE encoder. Tab. 18 shows the correlations of different reward models with aesthetics and text-image alignment. Aes-Corr is employed to assess the correlation with aesthetics, while the CLIP-Corr and VQA-Corr are utilized to measure alignment correlation. The calculation method for VQA-Corr is similar to that of CLIP-Corr, as described in Sec. 5.1, with the CLIP Score replaced by VQAScore. It is observed that the LRMs, especially LRM-XL, exhibit aesthetic and alignment correlations comparable to those of VLM-based models, as indicated by similar Aes-Corr and VQA-Corr values. The CLIP-Corr of LRMs is slightly lower than the average value of VLM-based reward models, likely because VLM-based models are fine-tuned versions of CLIP [9] or BLIP [19], leading to higher similarity to CLIP. The accuracy on validation and test sets

Table 18: The correlations of reward models with aesthetics and text-image alignment, along with the preference prediction accuracy on the validation and test set of Pick-a-Pic v1.

| | Model | Aes-Corr | CLIP-Corr | VQA-Corr | Val-Test Accuracy |
|---|---|---|---|---|---|
| Specific Reward Model (VLM-Based) | Aesthetic [38] | - | - | - | 54.03 |
| | CLIP Score (CLIP-H) [9] | - | - | - | 61.84 |
| | VQAScore [39] | - | - | - | 59.16 |
| General Reward Model (VLM-Based) | ImageReward [18] | 0.108 | 0.425 | 0.417 | 62.66 |
| | HPSv2 [17] | 0.007 | 0.602 | 0.406 | 64.76 |
| | HPSv2.1 [17] | 0.191 | 0.432 | 0.332 | 65.58 |
| | PickScore [16] | 0.066 | 0.490 | 0.402 | **71.93** |
| | Average | 0.093 | 0.487 | 0.389 | - |
| General Reward Model (Diffusion-Based) | LRM-1.5 ($t = 0$) | 0.115 | 0.359 | 0.339 | 65.46 |
| | LRM-1.5 ($t = 200$) | 0.111 | 0.356 | 0.336 | 67.21 |
| | LRM-XL ($t = 0$) | 0.073 | 0.403 | 0.390 | 67.44 |
| | LRM-XL ($t = 200$) | 0.089 | 0.401 | 0.385 | 69.31 |

Table 19: Comparison between SPO and LPO without MPCF. The general and aesthetic scores are calculated on the Pick-a-Pic validation unique set. If not specified, we use 20 inference steps.

| Model | Method | PickScore | ImageReward | HPSv2 | HPSv2.1 | Aesthetic | GenEval (20 Step) | GenEval (50 Step) |
|---|---|---|---|---|---|---|---|---|
| SD1.5 [14] | SPO | 21.22 | 0.1678 | 26.73 | 25.83 | **5.927** | 40.46 | 41.53 |
| | LPO wo MPCF | **21.49** | **0.4406** | **27.39** | **27.61** | 5.772 | **45.66** | **46.21** |

Table 20: Comparison results on T2I-CompBench++ [41] with 20 inference steps between SPO and LPO without MPCF.

| Model | Method | Color | Shape | Texture | 2D-Spatial | 3D-Spatial | Numeracy | Non-Spatial | Complex |
|---|---|---|---|---|---|---|---|---|---|
| SD1.5 [14] | SPO | 0.4112 | **0.4019** | 0.4044 | 0.1301 | 0.2909 | 0.4372 | 0.3008 | 0.2988 |
| | LPO | **0.4717** | 0.4002 | **0.4846** | **0.1793** | **0.3460** | **0.4596** | **0.3094** | **0.3171** |

of Pick-a-Pic is also present in Tab. 18. LRMs exhibit competitive performance, with larger variants (LRM-XL) achieving higher accuracy. Moreover, we observe that adding slight noise to the latent images ($t = 200$) leads to improved accuracy, which may be attributed to the fact that slight noise can better activate the feature extraction capabilities and improve the generalization of diffusion models.

## C  Additional Ablation Experiments

**Comparison between SPO and LPO without MPCF.** We conduct a comparison between SPO and LPO on identical training data. Specifically, we compare SPO with LPO without the MPCF strategy. As illustrated in Tab. 19 and Tab. 20, LPO without MPCF still outperforms SPO, especially in general and text-image alignment preference, demonstrating the effectiveness of reward modeling and preference optimization in the noisy latent space.

**The Compatibility with High Noise at Large Timesteps.** In Tab. 21, we analyze the compatibility of SPO and LPO with large training timesteps and high noises. Notably, for SPO, the training efficiency within the timestep range [0, 950] is significantly lower than that at [0, 750], and it is even inferior to the original model. In contrast, LPO optimization with this range remarkably enhances the model's performance. Furthermore, training only with the range [750, 950] can even achieve comparable performance to that of the range [0, 950], illustrating the superior adaptability of LRM to high-intensity noise at large timesteps.

**LRM's Sensitivity to Timesteps.** In Tab. 22, we investigate the timestep sensitivity of LRM. When LRM is consistently provided with a fixed timestep of 0 during LPO sampling, the model's performance degrades significantly. This highlights the importance of timestep input and confirms that LRM effectively leverages timestep information to predict preferences for noisy latent images.

**Regularization HyperParameter $\beta$.** $\beta$ is a regularization hyperparameter to control the deviation of the optimized model ($p_\theta$ in Eqn. (6)) with the reference model $p_{ref}$. As shown in Tab. 23, as $\beta$ increases, the regularization effect becomes stronger, which slows down the model's optimization

Table 21: Comparison between SPO and LPO with different training timestep ranges. We use 20 inference steps. * denotes the results quoted from [8].

| Model | Method | Range of $t$ | PickScore | ImageReward | HPSv2 | HPSv2.1 | Aesthetic | GenEval |
|---|---|---|---|---|---|---|---|---|
| | original | - | 20.56 | 0.0076 | 26.46 | 24.05 | 5.468 | 42.56 |
| | SPO* | [0, 750] | **21.43** | **0.1712** | **26.45** | - | **5.887** | - |
| SD1.5 [14] | | [0, 950] | 19.77 | -0.4529 | 22.72 | - | 5.111 | |
| | | [0, 700] | 21.02 | 0.3087 | 27.10 | 26.25 | 5.765 | 44.93 |
| | LPO | [750, 950] | 21.54 | 0.6337 | 27.47 | 27.64 | 5.853 | 48.28 |
| | | [0, 950] | **21.69** | **0.6588** | **27.64** | **27.86** | **5.945** | **48.39** |

Table 22: Comparison of timestep inputs of LRM during LPO sampling. We use 20 inference steps.

| Model | Input Timestep | PickScore | ImageReward | HPSv2 | HPSv2.1 | Aesthetic | GenEval |
|---|---|---|---|---|---|---|---|
| SD1.5 [14] | Real Timestep | **21.69** | **0.6588** | **27.64** | **27.86** | **5.945** | **48.39** |
| | 0 | 19.64 | -0.5043 | 25.37 | 21.74 | 5.283 | 32.21 |

Table 23: Ablation results on the hyperparamter $\beta$.

| $\beta$ | Aesthetic | GenEval | PickScore | ImageReward | HPSv2 | HPSv2.1 |
|---|---|---|---|---|---|---|
| 20 | 5.920 | 46.95 | 21.53 | 0.6407 | 27.54 | **28.13** |
| 100 | **6.031** | 48.23 | 21.68 | 0.6443 | 27.47 | 27.52 |
| 500 | 5.945 | 48.39 | **21.69** | **0.6588** | **27.64** | 27.86 |
| 1000 | 5.858 | **48.44** | 21.39 | 0.4785 | 27.22 | 26.47 |
| 5000 | 5.647 | 43.87 | 20.95 | 0.2970 | 27.00 | 25.68 |

speed and leads to poorer performance. Conversely, if $\beta$ is too small ($\beta = 20$), the regularization constraint becomes too weak, potentially harming the model's generalization and reducing image quality. The performance is optimal when $\beta$ is within the range of 100 to 500.

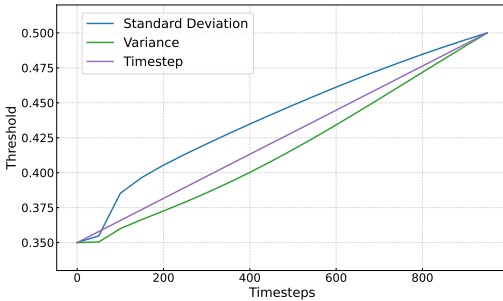

Figure 7: The thresholds of different strategies.

Figure 8: The number of valid samples of different timesteps .

**Other Dynamic Threshold Strategy.** In the main paper, we dynamically correlate the threshold in LPO sampling with the standard deviation $\sigma_t$ in DDPM [44]. Here we explore alternative strategies. The first strategy involves replacing $\sigma_t$ with variance $\sigma_t^2$. Consequently, the threshold can be formulated as:

$$th_t = \frac{\sigma_t^2 - \sigma_{min}^2}{\sigma_{max}^2 - \sigma_{min}^2} * (th_{max} - th_{min}) + th_{min}. \tag{16}$$

The second strategy is linearly correlating the threshold with the timestep $t$:

$$th_t = \frac{t - t_{min}}{t_{max} - t_{min}} * (th_{max} - th_{min}) + th_{min}, \tag{17}$$

where $t_{min}$ and $t_{max}$ denote the minimum and maximum optimization timesteps, respectively. The values of different strategies are illustrated in Fig. 7. The threshold range is $[0.35, 0.5]$ and the timestep range is $[0, 950]$. It can be observed that the standard deviation strategy sets higher thresholds for middle timesteps, while the variance strategy sets lower thresholds. As shown in Tab. 24, the standard deviation strategy achieves the best performance across various metrics. This

Table 24: Ablation results on the dynamic threshold strategies.

| Strategy | Aesthetic | GenEval | PickScore | ImageReward | HPSv2 | HPSv2.1 |
|---|---|---|---|---|---|---|
| Standard Deviation | **5.945** | **48.39** | **21.69** | **0.6588** | **27.64** | **27.86** |
| Variance | 5.921 | 47.78 | 21.57 | 0.5054 | 27.42 | 26.93 |
| Timestep | 5.929 | 48.22 | 21.65 | 0.6465 | 27.42 | 27.25 |

Table 25: Complete results on GenEval [40] with 20 inference steps.

| Model | Method | Single Object | Two Object | Counting | Colors | Position | Color Attribution | Overall |
|---|---|---|---|---|---|---|---|---|
| SD1.5 [14] | Original | 97.50 | 37.12 | 34.69 | 75.53 | 3.75 | 6.75 | 42.56 |
| | Diff.-DPO [4] | **98.44** | 38.38 | 36.25 | 77.93 | 4.50 | 7.25 | 43.79 |
| | SPO [8] | 95.00 | 33.84 | 32.50 | 69.95 | 4.25 | 7.25 | 40.46 |
| | SePPO [30] | 99.06 | 39.90 | 34.69 | **81.91** | 6.25 | 8.00 | 44.97 |
| | LPO | 97.81 | **54.80** | **40.94** | 79.52 | **7.00** | **10.25** | **48.39** |
| SDXL [11] | Original | 93.75 | 63.38 | 30.94 | 80.05 | 9.25 | 19.00 | 49.40 |
| | Diff.-DPO [4] | 99.06 | 76.52 | **45.00** | 88.83 | 11.50 | 25.75 | 57.78 |
| | MaPO [29] | 95.63 | 68.94 | 32.19 | 83.51 | 11.50 | 17.75 | 51.59 |
| | SPO [8] | 94.38 | 69.44 | 31.88 | 81.65 | 10.25 | 15.50 | 50.52 |
| | InterComp [31] | 99.06 | **82.83** | 43.75 | 84.04 | **14.00** | **31.75** | 59.24 |
| | LPO | **99.69** | 81.57 | 43.75 | **89.10** | **14.00** | 27.50 | **59.27** |

Table 26: Complete results on GenEval [40] with 50 inference steps.

| Model | Method | Single Object | Two Object | Counting | Colors | Position | Color Attribution | Overall |
|---|---|---|---|---|---|---|---|---|
| SD1.5 [14] | Original | **98.13** | 37.88 | 33.13 | 76.33 | 3.75 | 6.00 | 42.53 |
| | Diff.-DPO [4] | **98.13** | 41.16 | 37.81 | **81.91** | 4.50 | 7.25 | 45.13 |
| | SPO [8] | 95.63 | 36.62 | 34.83 | 72.34 | 3.75 | 6.50 | 41.53 |
| | SePPO [30] | **98.13** | 41.16 | 33.44 | 79.52 | **7.25** | 7.25 | 44.46 |
| | LPO | 97.81 | **55.30** | **42.19** | 80.59 | 6.75 | **10.00** | **48.77** |
| SDXL [11] | Original | 94.38 | 67.68 | 41.56 | 81.65 | 10.50 | 18.00 | 52.29 |
| | Diff.-DPO [4] | 99.06 | 77.78 | **49.69** | 86.17 | 13.25 | 27.50 | 58.91 |
| | MaPO [29] | 96.56 | 66.41 | 40.00 | 84.31 | 10.75 | 18.75 | 52.80 |
| | SPO* | 97.81 | 73.48 | 41.25 | 85.64 | 13.00 | 20.00 | 55.20 |
| | SPO [8] | 96.88 | 69.70 | 37.19 | 83.51 | 9.50 | 19.75 | 52.75 |
| | InterComp [31] | 99.06 | **85.10** | 41.25 | 86.97 | 13.50 | **32.00** | 59.64 |
| | LPO | **99.69** | 84.34 | 43.13 | **90.43** | **13.75** | 27.75 | **59.85** |

may be because the standard deviation strategy better accommodates the variations between $x_t$ at different timesteps.

Fig. 8 illustrates the number of valid samples of different timesteps for both constant and dynamic (standard deviation) threshold strategies. The dynamic threshold strategy significantly increases the number of valid samples, especially at intermediate timesteps.

**Complete Results on GenEval.** The complete GenEval results corresponding with that in Tab. 2 are presented in Tab. 25 and Tab. 26.

## D    Additional Visualizations

**Predicted Images $I_t$ at Different Timesteps.** Fig. 9 illustrates the predicted images $I_t$ generated through the diffusion forward and VAE decoding processes, as shown in Fig. 4 (a). As discussed in the main paper, predicted images at large timesteps tend to be very blurred, causing a significant distribution shift from the original images. This makes it challenging for PRMs to adapt to these blurred images without adequate pre-training and extensive datasets, resulting in unreliable predictions at large timesteps. In contrast, LRM can naturally perceive noisy latent images even at large timesteps, as it does during pre-training, representing a significant advantage over PRMs.

**Optimization Timesteps.** The generated images of different optimization timestep ranges are illustrated in Fig. 10. Larger timestep ranges result in more pronounced improvements in the quality of the generated images. We think there are two main reasons. Firstly, small timesteps in the denoising process primarily focus on high-frequency details, which do not lead to significant changes

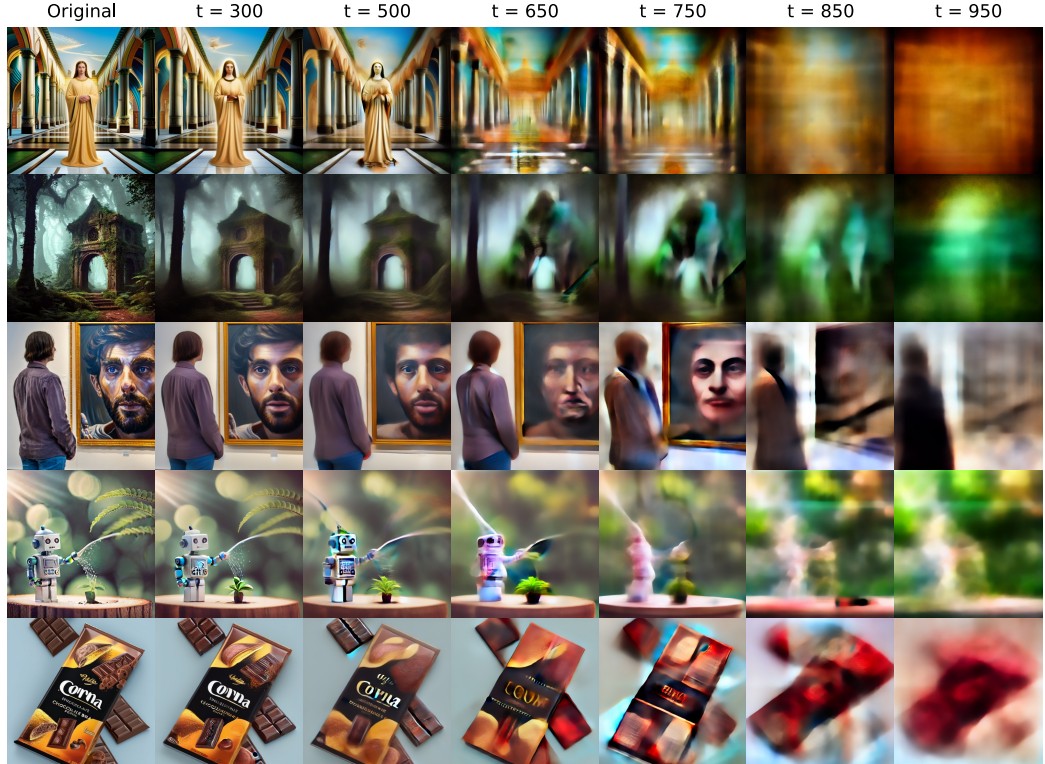

Figure 9: Predicted images $I_t$ in Fig. 4 (a) at different timesteps. The original images come from the Pick-a-Pic v1 dataset [16].

in the layout and style of the image. Secondly, as indicated in Fig. 8, the smaller the timestep, the fewer the valid samples, resulting in less optimization at the corresponding timesteps. Furthermore, compared to images in the range $[750, 950]$, those in $[0, 950]$ exhibit richer details, including both foreground and background, which also demonstrates that optimization at smaller timesteps aids in the enhancement of image details.

**More Comparison.** Fig. 11 presents the generated images of different methods based on SD1.5 [14]. Fig. 12 and Fig. 13 show the larger version of images in Fig. 1 with corresponding prompts. Fig. 14 and Fig. 15 provide more generated images of various methods based on SDXL [11]. Some keywords in prompts that other models fail to depict are highlighted using **bold** formatting.

# E    Additional Discussions

During the research process, we observed the following types of failure cases:

- Reward hacking: When training steps are excessive, the model exhibits reward hacking, where the reward metric continues to improve while the quality of generated images degrades. This stems from a certain misalignment between the reward model and human preferences, which remains a common issue across most existing methods. Mitigating this problem requires a combination of strategies, including improvements in preference data and reward modeling approaches.

- Decline in diversity: Extended training also leads to a reduction in generation diversity. This is caused by the reward model narrowing the output distribution of the diffusion model. A general solution is prompt engineering, such as rephrasing input prompts to enhance the diversity of generated images via prompt variation.

- Lack of fine details on SDXL: For some prompts, SDXL models optimized with LPO generate images with less detailed content compared to those optimized with SPO. This phenomenon was not observed on SD1.5. We will investigate this in future work to identify the underlying causes.

# F   Broader Impacts

This work introduces a preference optimization method for text-to-image diffusion models. The method may have the following impacts:

- The reward model plays a crucial role in shaping the preferences of diffusion models. However, if the training data for the reward model contains biases, the optimized model may inherit or even amplify these biases, leading to the generation of stereotypical or discriminatory content about certain groups. To mitigate this risk, it is essential to ensure that the training data is diverse, representative, and fair.

- This method offers a way to enhance the quality of generated images in terms of aesthetics, relevance, and other aspects by leveraging a well-designed reward model. It can also be applied to safety domains, optimizing the model to prevent the generation of negative content.

- The optimized model can generate highly realistic images, which may be used to create misinformation or misleading content, thereby impacting public opinion and social trust. Therefore, it is essential to develop effective detection tools and mechanisms to identify synthetic content.

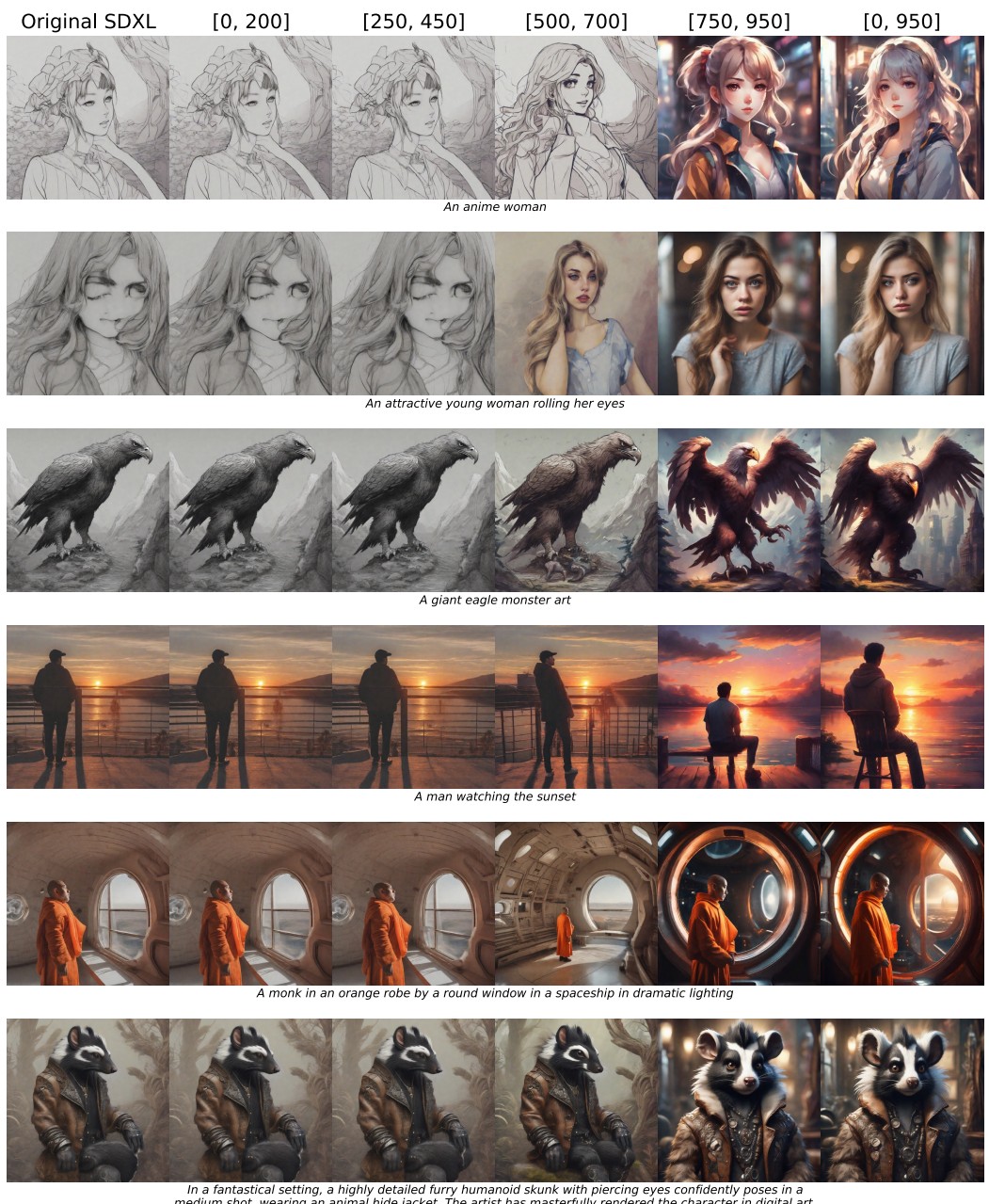

Figure 10: Qualitative comparison of various optimization timestep ranges based on SDXL.

| Original SD1.5 | Diffusion-DPO | SPO | LPO |
|:---:|:---:|:---:|:---:|

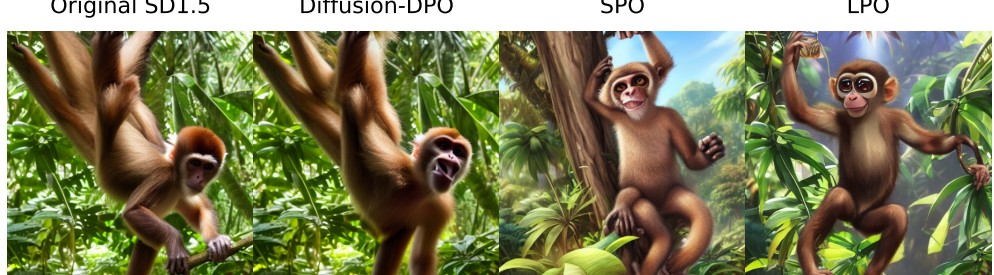

*a brown monkey with a **playful expression wears a pair of aviator sunglasses** as it swings through a lush jungle canopy. its tail curls around a vine, and the **sunlight reflects off its sunglasses**, capturing a sense of **adventure and fun**.*

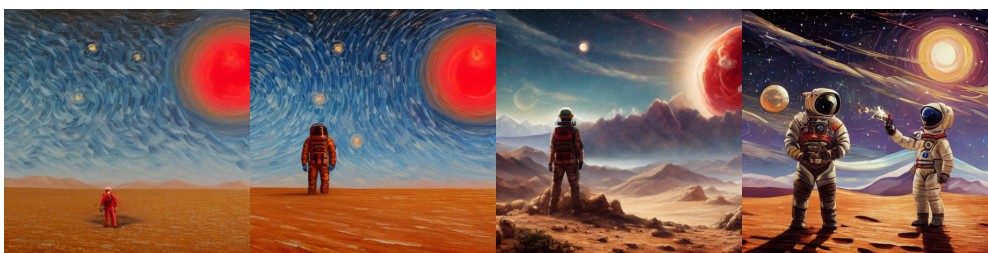

*an oil painting depicts a **starry** night in the desert, with a red **astronaut staring** intently at a blue **astronaut** who clutches a knife. the vast desert stretches out before them, dunes rippling in the moonlight. the sky above is **filled with twinkling stars**, while the distant horizon glows faintly from the setting sun. the tension between the **two astronauts** is palpable, as if a dramatic confrontation is about to unfold.*

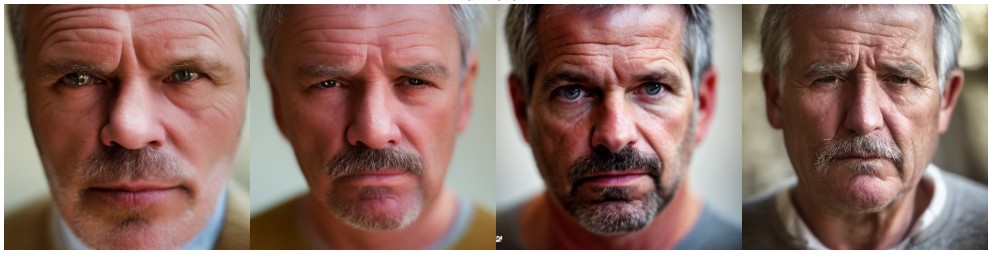

*Close-up portrait of middle-aged man, illuminated by soft natural light. His face is **deeply lined** and his eyes show emotion, **expressing sadness and contemplation**. Shot using an 85mm lens, the focus is on his facial features while **the background is softly blurred**. Soft lighting creates subtle highlights and shadows that accentuate the skin's texture.*

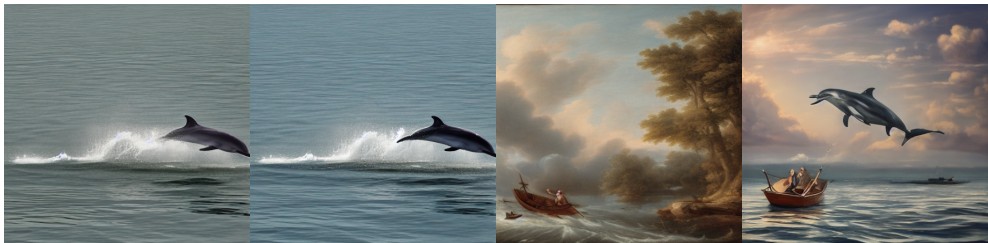

*A **dolphin** jumping over a **rowboat***

Figure 11: Qualitative comparison of various preference optimization methods based on SD1.5.

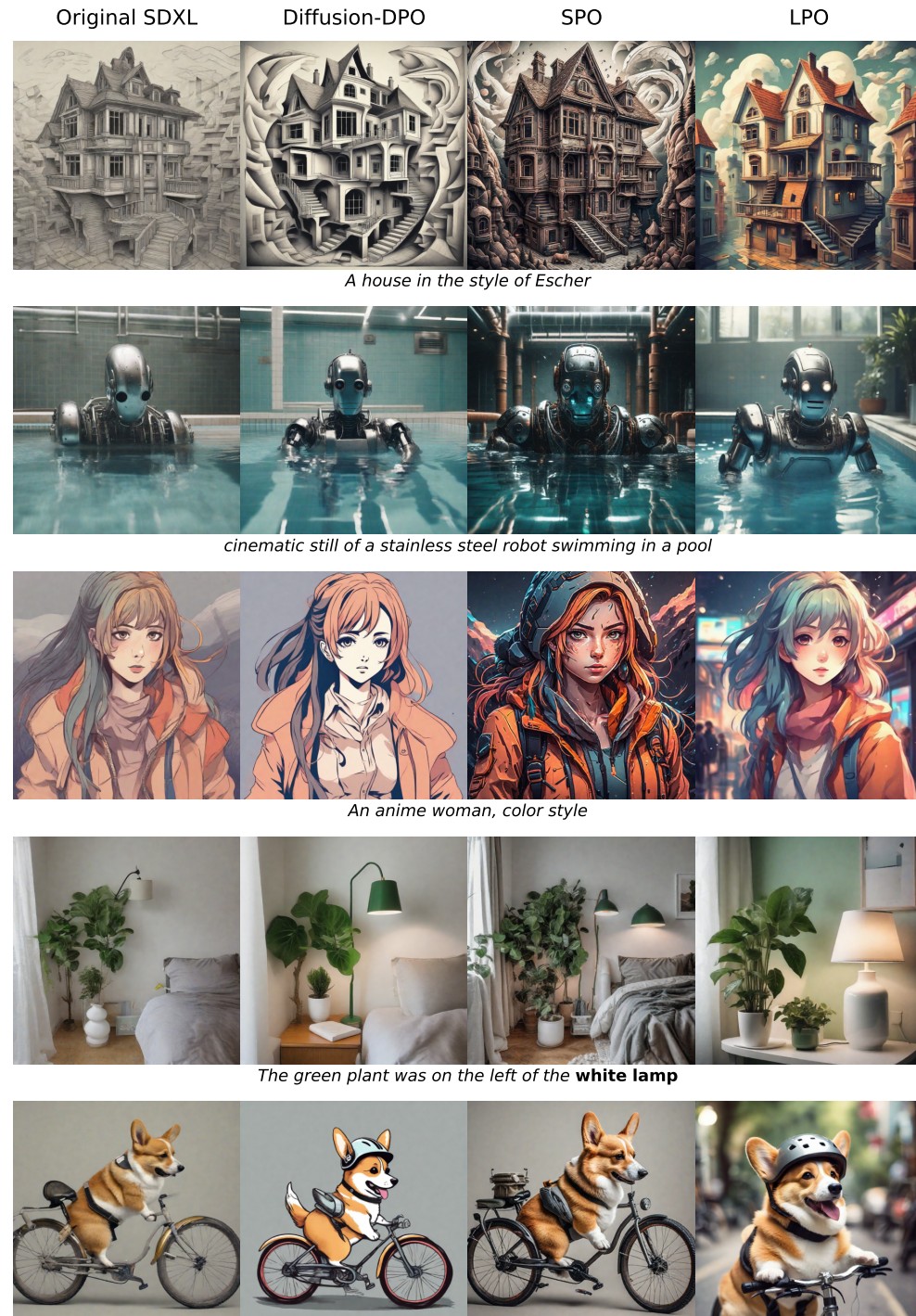

Figure 12: Qualitative comparison of various preference optimization methods based on SDXL.

| Original SDXL | Diffusion-DPO | SPO | LPO |

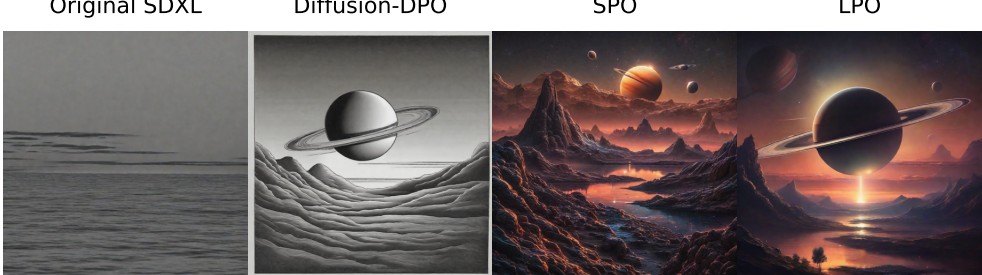

*Saturn **rises on the horizon***

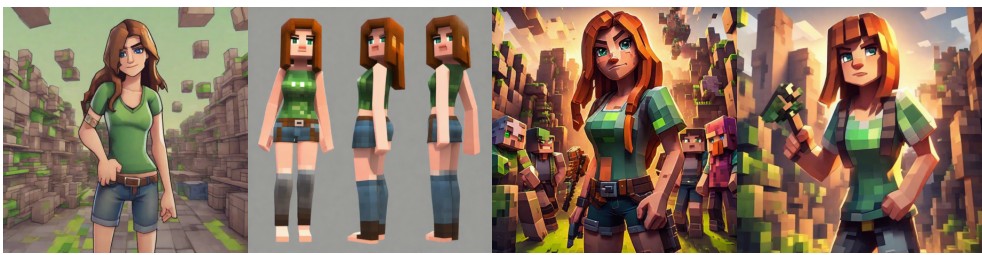

*A hot female Alex from **Minecraft***

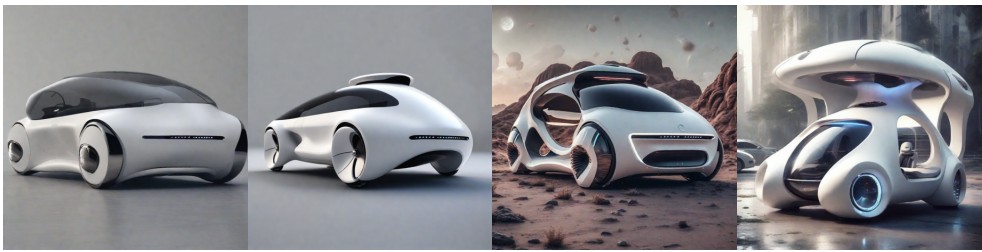

***Mushroom** concept car, **mushroom shape**, alien technology, interstellar transportation, cutting-edge technology, plasma core, white color*

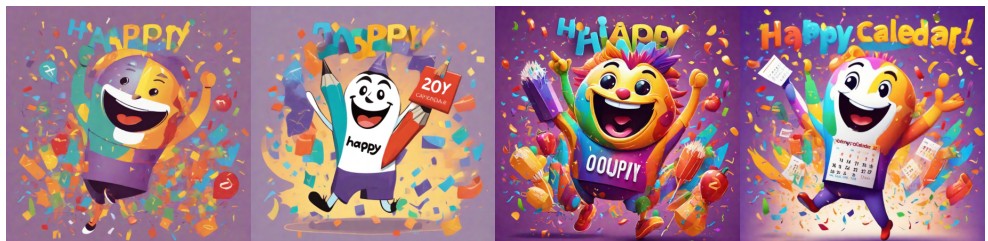

*a vibrant, smiling calendar mascot leaps in the air with glee, holding a pen and marking off days. the mascot is **surrounded by a confetti explosion of dates and months**, alluding to the joy and excitement of scheduling events. above the mascot, a bold, elegant **logo reads happy calendar** in a friendly, modern font.*

Figure 13: Qualitative comparison of various preference optimization methods based on SDXL.

| Original SDXL | Diffusion-DPO | SPO | LPO |
|---|---|---|---|

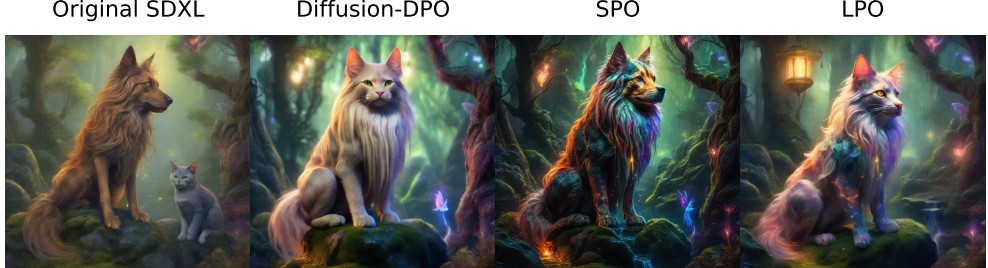

*a majestic dog-cat chimaera is perched on a moss-covered rock in an enchanted forest, with **its feline face and canine body seamlessly blended**. **it sports a luxurious mane and bushy tail**, while vibrant, ethereal lights dance around it, enhancing the digital fantasy atmosphere.*

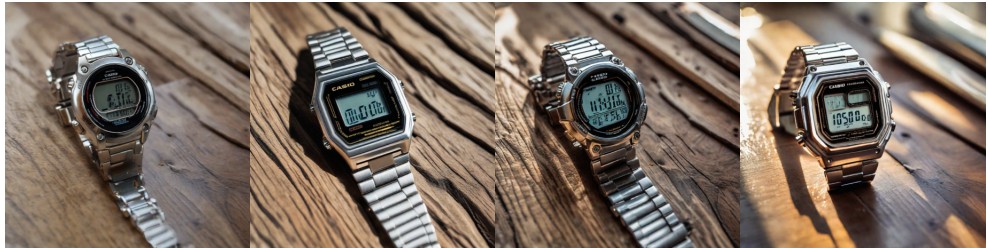

*a sleek, silver casio watch with a digital display sits atop a polished wooden surface. its metallic band catches the sunlight streaming through a nearby window, **casting a warm glow on the watch face**. the time reads 12:34 pm, while the stopwatch feature ticks away in the background.*

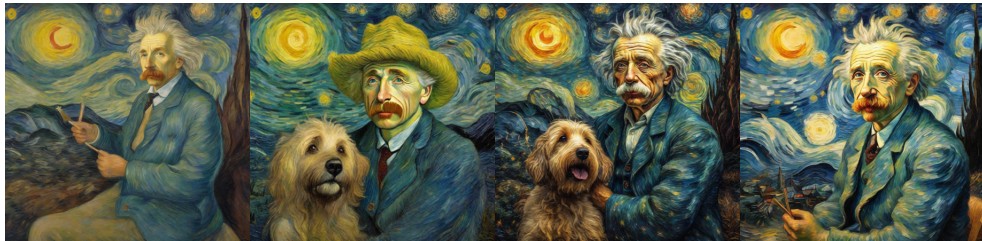

*Einstein's famous photo, Van Gogh's Starry Night style, the oil painting replaces the characters in Van Gogh's self-portrait with Einstein's portrait, a masterpiece*

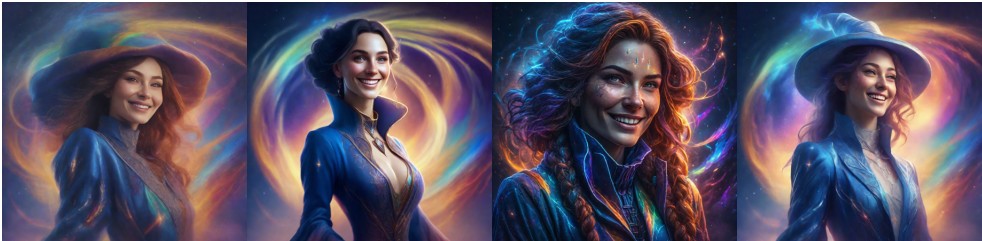

*A smiling beautiful sorceress wearing a high necked blue suit **surrounded by swirling rainbow aurora**, hyper-realistic, cinematic, post-production*

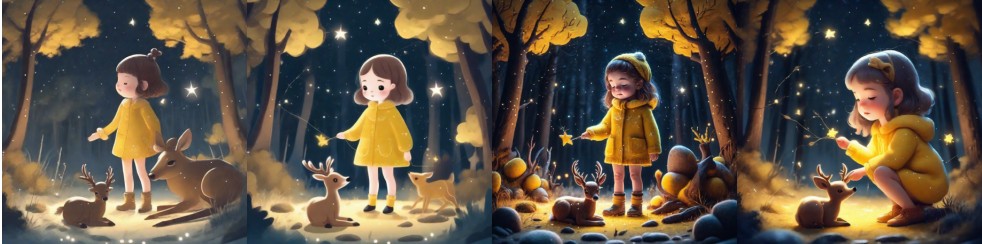

*Wool felt texture, at night, under the starry sky, in the forest, a little girl wearing yellow clothes, **with cute hairpins**, **squats on the ground**, **stroking a deer**, tiny particles glow, perfect composition,*

Figure 14: Qualitative comparison of various preference optimization methods based on SDXL.

| Original SDXL | Diffusion-DPO | SPO | LPO |
|---|---|---|---|

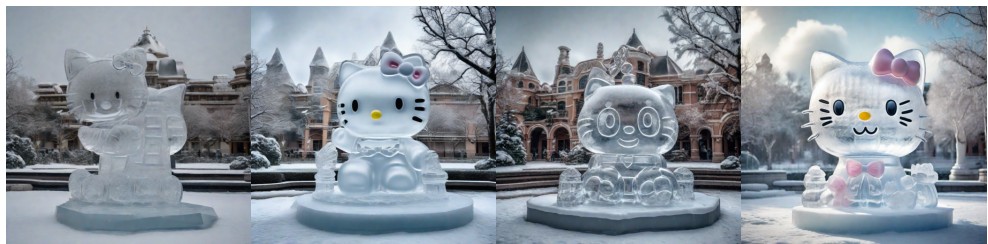

*Giant Hello Kitty ice sculpture, crystal clear ice sculpture art, sitting posture showing **the iconic bow**, classic round face design including beard and round nose, finely carved fish patterns on the dress, and **transparent geometric shape** to create the image of the beloved character. Stepped ice base, **winter park environment background**, ground covered with snow, cloudy day with diffuse light, figures in front showing size proportions, transparent ice showing internal light refraction, professional photography perspective, ultra-high definition image quality, magnificent scale, Exquisite ice sculpture craftsmanship, winter festival atmosphere*

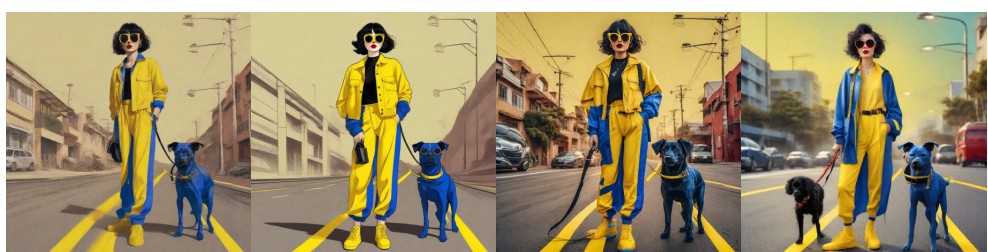

*A woman wearing an **ultramarine jacket and yellow slacks**, yellow socks and yellow shoes, wearing all black sunglasses, **short wavy hair without bangs**, holding **a black dog** with a blue leash in her hands, the dog in her Right, fashion model, red lipstick, on an asphalt road with a yellow line, single yellow background, oriental fusion clothing, low angle, like a fashion pictorial, concept art, decorative, hyper-detailed, realistic, bright colors, realistic, model posture*

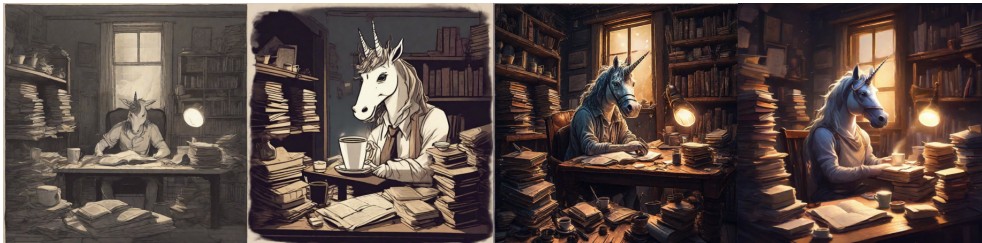

*a weary unicorn grad student sits at a cluttered wooden desk, surrounded by stacks of books and papers. the dim light from a small desk lamp illuminates the room, casting shadows on the walls. dark circles are evident under the unicorn's eyes, **as it holds a cup of coffee in one hoof**, trying to stay awake while studying. the moonlight from the window reflects off its horn, adding a magical touch to the scene.*

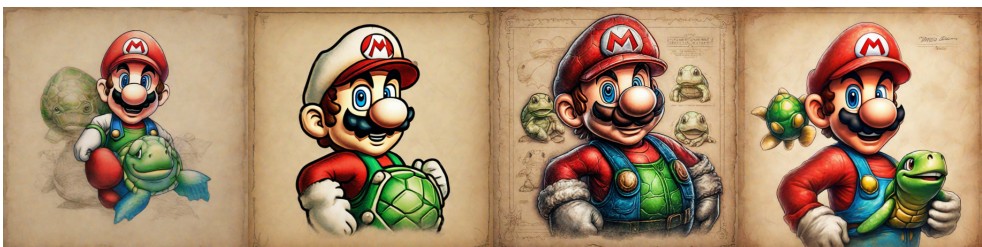

*a detailed sketch of mario from the classic video game series is depicted on a textured parchment. he wears his signature **red cap** and overalls, white gloves, and **holds a green turtle shell**. the drawing captures his **cheerful expression** and iconic mustache, with fine pencil strokes adding depth and dimension.*

Figure 15: Qualitative comparison of various preference optimization methods based on SDXL.

