# OpenReview forum: "Diffusion Model as a Noise-Aware Latent Reward Model for Step-Level Preference Optimization"
_NeurIPS.cc/2025/Conference — NeurIPS 2025 poster_

### Official Review · Reviewer_2Nf4 · 2025-06-30

**Clarity:** 2
**Significance:** 3
**Originality:** 2
**Rating:** 4
**Confidence:** 3

**Summary:**

This paper proposes LRM (Latent Reward Modeling) and LPO (Latent Preference optimization) as ways to improve the aesthetic and text alignment of the model. LRM allows for step-level preference optimization directly in the latent space. This is done by using a pretrained diffusion model as it already provides the characteristics for noise-aware, step-dependend modeling. Further the paper proposes VFE and MPCF to improve image-text correlation in UNet-Features and avoid the sometimes contradictory labels in the Pick-a-Pic dataset. The method shows promising results while being cheaper to use than their competitors.

**Questions:**

- Did you observe any failure cases like reward hacking?
- Could you provide more reasoning for MPCF (regarding W4) and motivation for it?

I am willing to increase my score if my concerns (esp. W1 / Clarity + W3 + W4) are well addressed by the authors.

**Ethical Concerns:**

["NO or VERY MINOR ethics concerns only"]

**Final Justification:**

The main concern for me was clarity. In the rebuttal the authors provided good explanations and promised a revision of certain aspects that will help make the work overall more structured. Therefore I raised my score.

**Limitations:**

The Limitations & Future Work section concentrates more on the latter and the only limitation discussed is the dataset. There are no limitations discussed that are relevant to the core method.

**Quality:**

3

**Strengths And Weaknesses:**

Weaknesses:
- The paper misses structure and especially a list of core contributions as the authors introduce / propose LRM, MPCV, LPO and VFE which I would consider core contributions.
- The VFE experiment (Table 8), shows that without VFE Aesthetic score and AesCorr is higher than with any VFE configuration which makes me doubt the general usefullness of VFE.
- The authors reason in l.181 - l. 188 that the winning and loosing $x_t$ are dependend on $t$ and the respective amount of noise added which is understandable. However, a supporting experiment that provides intuition is missing. This also comes back to W1, where MPCV is motivated using this argument of noise dependend $x^w_t$ and $x^l_t$ but there is no proof of this motivation in terms of references or own experiments.
- Extending the previous point, it is pointed out that int he Pick-a-Pic v1 dataset $x^w_t$ does not clearly perform better on eval benchmarks (Clip Score Aesthetic Score and VQAScore) than $x^l_t$. In the following paragraph (l. 196 onward), MPCF is explained which just says that instead of relying on a single of the benchmarks one takes multiple. This is a weak argument especially as the authors previously criticize the correlation between $x^w_t$ and these benchmarks one a few lines above.
- Following this reasoning, it would be vital to include the user study in the main paper.
- Some minor spelling mistakes (l. 150, "averagely"-> "average pooled")

Strengths:
- The core method (LRM) makes sense and is intuitive.
- It provides good result on the T2I-CompBecnh ++ which shows that this method can provide meaningful improvements compared to the baseline diffusion model and their competitors while requiring less training time.
- There are numerous ablations that show good settings which give more intuition of the method.

---

> ### Author Rebuttal · Authors · 2025-07-31
>
> Thanks for your effort in reviewing our paper and giving appreciation for our novelty, methods, and experiments. We hope the following responses solve your concerns.
>
> > Q1 (Weakness 1): The paper misses structure and especially a list of core contributions as the authors introduce / propose LRM, MPCV, LPO and VFE which I would consider core contributions.
>
> R1: Thanks for your constructive question. In the Introduction section, we did not explicitly list the core contributions of this work. We will add the following content in the revised version:
>
> The core contributions of this paper are summarized as follows:
>
> - A noise-aware Latent Reward Model (LRM) is introduced, which repurposes the pre-trained diffusion model for step-level reward modeling in the noisy latent space, with a Visual Feature Enhancement (VFE) module being used to enhance LRM's focus on the text-image alignment.
> - A Multi-Preference Consistent Filtering (MPCF) strategy is proposed to refine the public preference dataset, enabling LRM to better align with human preferences.
> - A Latent Preference Optimization (LPO) method based on LRM is introduced to perform step-level preference optimization directly within the noisy latent space of diffusion models.
> - Extensive experimental results demonstrate the effectiveness, efficiency, and generalization ability of the proposed methods.
>
> In the Method section, we introduce LRM, MPCF, and LPO in three separate subsections, following the order of "reward model architecture" → "reward model data filtering and training" → "optimization method". VFE is included in the LRM subsection. We provided a summary of this structure in Lines 130–133. We believe this ordering is reasonable, and we hope the revised version will better highlight our contributions and present a more coherent overall structure.
>
> > Q2 (Weakness 2): The VFE experiment (Table 8), shows that without VFE Aesthetic score and AesCorr is higher than with any VFE configuration which makes me doubt the general usefullness of VFE.
>
> R2: Thanks for your detailed question. **VFE serves as a trade-off module between alignment and aesthetics.** The VFE module is inspired by the Classifier-Free Guidance (CFG) [1]. Its purpose is to enhance the weight of text-related visual features within the overall visual representation (Equation 8), thereby increasing LRM’s focus on text-image alignment. However, this comes at the cost of reducing the contribution of visual features unrelated to the text, which leads to a decrease in the correlation between LRM and aesthetics (Aes-Corr).  Although a slight decline in the Aesthetic score, it significantly improves text-image alignment (GenEval) and the overall preference score (PickScore), as discussed in Lines 308–312.
>
> > Q3 (Weakness 3): The authors reason in l.181 - l. 188 that the winning and loosing $ x_t $ are dependend on t and the respective amount of noise added which is understandable. However, a supporting experiment that provides intuition is missing. This also comes back to W1, where MPCV is motivated using this argument of noise dependend $x_t^w$ and $x_t^l$ but there is no proof of this motivation in terms of references or own experiments.
>
> R3: Thanks for your constructive suggestion. We regret the limited explanation on this point. We respond in the following aspects:
>
> - **Analysis of this motivation**: What we intended to convey in the paper is that if $x_0^w$ outperforms $x_0^l$ only in certain aspects, then after adding noise of equal intensity during the forward process, $x_t^w$ may become worse than $x_t^l$. The dual problem in the reverse denoising process is: **for the same prompt, two generated images $I_a$ and $I_b$ may have inconsistent preferences at intermediate steps versus the final output—$x_t^{I_a}$ being better than $x_t^{I_b}$ does not necessarily imply $x_0^{I_a}$ is better than $x_0^{I_b}$.**
>
> - **Proof of this motivation**: The above dual problem has been discussed in prior work. For example, in ImageReward [2] (Figure 4), the authors run inference with different seeds for the same prompt and evaluate the ImageReward scores of intermediate steps $x_t$ ($\hat{x}\_{0,t}$). The curves corresponding to different seeds show many intersections, each indicating a change in preference order. **For instance, between the brown and pink lines, at step 25, the score of $x_{25}^{brown}$ is higher than $x_{25}^{pink}$, but at step 40, $x_{40}^{pink}$ surpasses $x_{40}^{brown}$, illustrating that preferences in the intermediate steps of diffusion models do not necessarily align with final-step preferences.** Similarly, Figure 1 in SPO [3] also visualizes this kind of inconsistency.
>
> We will incorporate these prior discussions in the revised version to provide a more comprehensive motivation for MPCF.
>
> > Q4 (Weakness 4): Extending the previous point, it is pointed out that in the Pick-a-Pic v1 dataset $x_t^w$ does not clearly perform better on eval benchmarks (Clip Score Aesthetic Score and VQAScore) than $x_t^l$. In the following paragraph (l. 196 onward), MPCF is explained which just says that instead of relying on a single of the benchmarks one takes multiple. This is a weak argument especially as the authors previously criticize the correlation between $x_t^w$ and these benchmarks one a few lines above.
>
> R4: Thanks for your detailed question. We'd like to emphasize that **we are not criticizing the reward models (benchmarks)  themselves—in fact, we rely on their robustness—but rather questioning the label quality of the Pick-a-Pic dataset**. As some of its original labels may not be trustworthy, we use the more reliable reward models to filter the data. We believe there may be some misunderstanding. To clarify, we outline the overall logic behind MPCF as follows:
>
> - First, we argue that if the training data for LRM is of low quality—specifically, if the winning image is only marginally better than the losing image in one particular aspect—then after injecting the same level of noise during the forward process, the resulting noisy version of the winning ($x_t^w$) may actually become worse than that of the losing ($x_t^l$). This motivates a stricter assumption: if the winning image is superior to the losing image in **multiple aspects**, then the relative preference between the two images is more likely to remain consistent after noise is added.
>
> - Consequently, during LRM training, we aim to use only image pairs where the winning is clearly better than the losing across multiple dimensions. However, this is not always the case with the Pick-a-Pic dataset [4]. Since it was constructed via crowdsourcing, some image pairs have low-quality labels, which means that the winning image may not be better in all aspects, as illustrated in Figure 5.
>
> - To filter out such unreliable training pairs in Pick-a-Pic, we leverage three open-source reward models: the Aesthetic Predictor for aesthetic judgment, and CLIPScore and VQAScore for text-image alignment evaluation. If the original label in Pick-a-Pic ($I^w$ vs. $I^l$) is consistent with all three reward models ($G=S(I^w)-S(I^l)>0$), we regard the winning image as being superior in multiple aspects and retain that pair for LRM training.
>
> We hope the above explanation helps clarify the motivation and rationality of MPCF. We will revise the manuscript accordingly to make it easier to follow.
>
> > Q5 (Weakness 5): Following this reasoning, it would be vital to include the user study in the main paper.
>
> R5: Thanks for your suggestion. If the user study refers to evaluating the LPO-optimized model, the results are shown in Figure 6 in the appendix. If it refers to the Pick-a-Pic dataset, conducting a user study for such a large dataset is costly, which is why MPCF uses three reliable reward models for automatic filtering. These models serve as proxies to approximate human judgments in a more scalable way. We’re willing to follow further suggestions if you have specific ideas.
>
> > Q6 (Weakness 6): Typos.
>
> R6: Thanks for your detailed review. We will correct the following typos in the revised version.
>
> - Line 150: "averagely" -> "average pooled"
> - Table 23: "Ablataion" -> "Ablation"
>
>
>
> > Q7 (Question 1): Failure cases like reward hacking?
>
> R7: Yes. When training steps are excessive, the model exhibits reward hacking, where the reward metric continues to improve while the quality of generated images degrades. This stems from a certain misalignment between the reward model and human preferences, which remains a common issue across most existing methods. Mitigating this problem requires improvements in training data and reward modeling approaches. We will include the discussion about failure cases in the revised version.
>
> > Q8 (Question 2): More reasoning for MPCF (regarding W4) and motivation for it?
>
> R8: Thanks for your constructive suggestions. We provide more details in **R3** and **R4**.
>
> > Q9 (Limitation 1): There are no limitations discussed that are relevant to the core method.
>
> R9: Thanks for your valuable suggestion. We will include the following limitations in the revised version:
>
> - Since LPO is performed in the latent space defined by the VAE encoder, the VAE used in the LRM must be identical to that in the diffusion model to be optimized.
>
> - MPCF relies on three reward models to approximate human preferences for automatic data filtering. However, the filtered data may still inherit biases or limitations shared by these reward models.
>
> [1] Ho J, Salimans T. Classifier-free diffusion guidance. arXiv preprint arXiv:2207.12598, 2022.
>
> [2] Xu J, Liu X, Wu Y, et al. Imagereward: Learning and evaluating human preferences for text-to-image generation. NeurIPS, 2023.
>
> [3] Liang Z, Yuan Y, Gu S, et al. Aesthetic post-training diffusion models from generic preferences with step-by-step preference optimization. CVPR 2025.
>
> [4] Kirstain Y, Polyak A, Singer U, et al. Pick-a-pic: An open dataset of user preferences for text-to-image generation. NeurIPS, 2023.

---

> > ### Author Response · Authors · 2025-08-04
> >
> > Dear Reviewer 2Nf4，
> >
> > Thank you again for your time and effort in reviewing our paper. As the discussion phase is drawing to a close, we would like to kindly ask whether our rebuttal has adequately addressed your concerns. If there are any remaining questions or points that need further clarification, we would be happy to provide additional details within the remaining time. If you feel that your concerns have been resolved, we would sincerely appreciate it if you would consider updating your score to reflect the newly added results and discussion. We would also greatly value the opportunity to continue the discussion if needed.

---

> ### Comment · Reviewer_2Nf4 · 2025-08-04
>
> Thank you for your rebuttal.
>
> My main concern for this work was clarity. The authors have delivered a clear and well written rebuttal and have explained most points i was confused about very well.
>
> My concerns are resolved and I will increase my score.

---

> > ### Author Response · Authors · 2025-08-04
> >
> > Dear Reviewer 2Nf4,
> >
> > Thank you very much for your thoughtful feedback and for the revised score. We truly appreciate your recognition of our work and your valuable insights, which have helped us improve the paper.
> >
> > Best regards!

---

### Official Review · Reviewer_rhE9 · 2025-07-01

**Clarity:** 3
**Significance:** 3
**Originality:** 3
**Rating:** 4
**Confidence:** 3

**Summary:**

The paper proposes a novel way to train latent reward models for improving diffusion models. It shows that the pretrained diffusion models are naturally suited for step-level reward modelling. The architecture design for the reward model is natural and novel. Ablations on the architectures also present the robustness of the design choice. Also, the final results on benchmarks on Pick Score, HPS, ImageReward, GenEval, etc., demonstrate the excellence of the LRM. Due to its latent nature, the inference efficiency is higher than in previous works.

**Questions:**

1. The reward modeling is compatible with the U-Net architecture, which is widely used. Future exploration could focus on transformer-based architectures. What is the main architectural difference between SD3 in the appendix and SDXL in the main paper?

2. How sensitive is LRM to the sampling method? There are many sampling strategies, and the proposed LPO is only suitable for some slower stochastic samplers. It is beneficial to examine the compatibility of LPO with different samplers.

3. The proposed LRM significantly reduces the computational cost of image evaluation. However, during inference, the diffusion model still needs to generate more samples in the latent space. A fair efficiency comparison with baseline methods would be helpful—for example, measuring the time required to obtain 100 good images after selection by the reward model.

**Ethical Concerns:**

["NO or VERY MINOR ethics concerns only"]

**Final Justification:**

the rebuttal has addressed some of my concerns. I thus maintain my score.

**Limitations:**

yes

**Quality:**

3

**Strengths And Weaknesses:**

Strengths:

1. The paper is easy to read and well-arranged.
2. The architecture design together with the ablation studies are insightful and clean.
3. The empirical results show the excellence of the proposed LRM and LPO.


Weaknesses:

1. The current reward model is restricted to U-Net architecture, which may not be compatible with SOTA diffusion transformer architecture.

---

> ### Author Rebuttal · Authors · 2025-07-31
>
> Thanks for your effort in reviewing our paper and giving appreciation for our writing, novelty, and methods. We hope the following responses solve your concerns.
>
> > Q1 (Weakness 1): The current reward model is restricted to U-Net architecture, which may not be compatible with SOTA diffusion transformer architecture.
>
> R1: Thanks for your concern. In the appendix (Lines 521–535, Tables 13 and 14), **we have verified the effectiveness of LRM and LPO on SD3‑Medium, which is a model based on the diffusion transformer (DiT) architecture.**
>
> **In addition, we also conduct experiments on two first‑tier, internal DiT‑based models.** We compare two different reward models: one is LRM, which utilizes part of the parameters of the DiT, and the other is a VLM with more than 25B parameters. They are trained on internal image‑text alignment datasets and then employed for the preference optimization of the DiT models. For the same diffusion model, we train two reward models using identical data. The human‑evaluation results for image‑text alignment of optimized DiT models are as follows:
>
> | Diffusion Model | Reward Model | Alignment |
> | --------------- | ------------ | --------- |
> | DiT-S       | LRM (9.5B)   | **72.5**  |
> | DiT-S       | VLM (>25B)   | 71.2      |
> | DiT-L         | LRM (15B)    | **82.8**  |
> | DiT-L        | VLM (>25B)   | 80.8      |
>
> All the above experimental results demonstrate that our methods can be effectively applied to the DiT architecture and outperform larger VLMs.
>
>
>
> > Q2 (Question 1): The reward modeling is compatible with the U-Net architecture, which is widely used. Future exploration could focus on transformer-based architectures. What is the main architectural difference between SD3 in the appendix and SDXL in the main paper?
>
> R2: Thanks for your question and constructive suggestion. We will continue to explore transformer‑based architectures. The SD3 model [1] in the appendix is based on the multimodal diffusion transformer (MMDiT) architecture, where text and image features interact through a self‑attention mechanism. In contrast, the SDXL model [2]  in the main paper is based on the U‑Net architecture, where text and visual features interact through cross‑attention.
>
>
>
> > Q3 (Question 2): How sensitive is LRM to the sampling method? There are many sampling strategies, and the proposed LPO is only suitable for some slower stochastic samplers. It is beneficial to examine the compatibility of LPO with different samplers.
>
> R3: Thanks for your detailed question. We believe there may be some misunderstanding here. LRM and LPO are not only suitable for slower stochastic samplers. We address this from the following two perspectives:
>
> - **Training of diffusion models**: LPO is an online-sampling-based method that directly optimizes the parameters of the diffusion model, rather than a training-free test-time scaling (TTS) approach. As such, LPO requires an SDE-form stochastic sampler for exploration and sampling, **but it is not limited to slow stochastic samplers.** In fact, in our experiments on SD1.5 and SDXL, we used a **20-step DDIM sampler** (Equations 2, 3, and 4) during LPO sampling (Figure 4 (d)), with $\eta$ set to 1 to convert it into the SDE form. In the SD3 experiments in the appendix, we switched Flow Matching from ODE to SDE (Equations 14 and 15) and also used **20 steps** for LPO sampling. **In topics that do not focus on sampling acceleration, we believe that a 20-step stochastic sampler is not slow.**
> - **Inference of diffusion models**: During inference evaluation of diffusion models optimized by LPO, we use deterministic samplers in ODE form. Specifically, for SD1.5 and SDXL, we use a **20-step DDIM sampler** with $\eta$ set to 0 (Tables 2 and 3), and for SD3, we use a **20-step Euler sampler** (Tables 13 and 14).
>
> These experimental settings effectively demonstrate that both the LRM and LPO methods, as well as diffusion models optimized by LPO, can consistently improve performance across a variety of samplers. We believe the misunderstanding may be caused by the lack of detailed descriptions of the samplers used in the paper. We will provide a more detailed explanation of the sampling strategies in the revised version.
>
>
>
> > Q4 (Question 3): The proposed LRM significantly reduces the computational cost of image evaluation. However, during inference, the diffusion model still needs to generate more samples in the latent space. A fair efficiency comparison with baseline methods would be helpful—for example, measuring the time required to obtain 100 good images after selection by the reward model.
>
> R4: Thank you for recognizing the efficiency of LRM and for your valuable suggestion. Regarding inference, there may be some misunderstanding. **LPO is an online-sampling-based method that directly optimizes the parameters of the diffusion model, rather than a training-free test-time scaling (TTS) approach.**
>
> - If "inference" refers to the sampling procedure during LPO, it is indeed the case that multiple $x_t$ samples are drawn at each step and scored by the reward model to construct win–lose pairs for LPO training. This exploration process is similar to other online optimization methods such as SPO [3]. Since each $x_t$ is sampled from a Gaussian distribution $p(x_t|x_{t+1})$, the sampling itself requires negligible cost. The main computational overhead lies in scoring multiple $x_t$ samples with the reward model. **In this regard, LRM has a significant advantage over VLMs, as it avoids both $\hat{x}_{0,t}$ prediction and VAE decoding (Figure 4 (a) and (b)). We provide a fair efficiency comparison in Table 5 of the paper.**
>
> - If "inference" refers to the inference of the optimized model after LPO training, **we do not generate multiple samples in latent space. Instead, we use a deterministic ODE-based sampler directly for inference**.
>
> Regarding your suggestion—"measuring the time required to obtain 100 good images after selection by the reward model"—we are somewhat confused, as this seems to be an evaluation strategy typically used in training-free methods. If possible, we would appreciate it if you could provide more details on how such a comparison could be conducted in our setting. We would be glad to perform this experiment if applicable.
>
>
>
> [1] Esser P, Kulal S, Blattmann A, et al. Scaling rectified flow transformers for high-resolution image synthesis. ICML, 2024.
>
> [2] Podell D, English Z, Lacey K, et al. Sdxl: Improving latent diffusion models for high-resolution image synthesis. ICLR, 2024.
>
> [3] Liang Z, Yuan Y, Gu S, et al. Aesthetic post-training diffusion models from generic preferences with step-by-step preference optimization. CVPR 2025.

---

> > ### Author Response · Authors · 2025-08-04
> >
> > Dear Reviewer rhE9,
> >
> > Thank you again for your time and effort in reviewing our paper. As the discussion phase is nearing its end, we kindly ask whether our rebuttal has sufficiently addressed your concerns. If there are any remaining questions or points requiring clarification, we would be happy to provide further details within the remaining time. We would greatly appreciate the opportunity to engage in further discussion with you.

---

> > > ### Comment · Reviewer_rhE9 · 2025-08-05
> > >
> > > I appreciate the authors' efforts in the rebuttal. Thanks for your clarification on the inference time compute. I thus would like to maintain my original rating.

---

> > > > ### Author Response · Authors · 2025-08-06
> > > >
> > > > Dear Reviewer rhE9,
> > > >
> > > > Thank you for your thoughtful feedback and positive assessment. We truly appreciate your recognition of our work and your valuable insights, which have helped us improve the paper.
> > > >
> > > > Best regards!

---

### Official Review · Reviewer_Xmnx · 2025-07-01

**Clarity:** 4
**Significance:** 3
**Originality:** 4
**Rating:** 5
**Confidence:** 5

**Summary:**

The paper proposes LPO (Latent Preference Optimization), a follow-up to the SPO approach for preference optimization and text-to-image alignment. Both methods focus on training a model to optimize during the denoising steps of the diffusion process. However, compared to SPO, which needs a pixel-level evaluator, LPO uses a Latent Reward Model (LRM) trained for alignment directly in the latent space without needing a decoder to extract the pixel representations.

- Training purely in latent space means no need for the decoding step and access to full noise range.
- They propose Visual Feature Enhancement (VFE) which is based on Classifier-Free Guidance to amplify text conditioning
- Multi-Preference Consistent Filtering (MPCF), which is used to counter many issues in the Pick-a-Pic dataset that is widely used but noisy and low quality.
- These changes result in faster training and SOTA results for alignment and aesthetics.

**Questions:**

- As stated in the weaknesses, the paper should introduce failure cases and specific examples where the current method struggles (even with the ideal hyperparmeters) and provide some insight as to why this happens. This is vital for future work and understanding the limitations.

- I have a few concerns with the speed comparison, which I think are very important to address. Looking at the SPO paper, they claim "We use 4× A100 GPUs, which take ∼12 and ∼29.5 hours to train SD-1.5 and SDXL, respectively." but also "We also spend 8 and 29 hours training SPM for SD-1.5 and SDXL, respectively."  Looking at the comparison SRM and LRM require extra training time so the comparison with DPO is not fair and extra training should be accounted for.

- I was also not able to find the DPO training time figures so I think a specific source should be included or if the authors reproduced it themselves they should provide details for this.

- While the ablation studies are extensive and well executed, they are conducted only on SD1.5.  However SDXL and SD1.5 can be very different. It will be beneficial to perform threshold, VFE and MPCF experiments on SDXL to show that hyper-parameters generalize across model scales.

**Ethical Concerns:**

["NO or VERY MINOR ethics concerns only"]

**Final Justification:**

The authors have addressed my concerns thus I am willing to maintain my score.

**Limitations:**

- Should add "Since LPO is performed within the latent space, which is determined by the VAE encoder, the VAE of LRM should be identical to that of DMO."

- The training process uses MPCF, which filters data based on scores like Aesthetic Score, CLIP Score and VQAScore. This process can inherit any issues that are shared across these metrics.

**Paper Formatting Concerns:**

Minor: Table 23: Ablataion -> Ablation

**Quality:**

4

**Strengths And Weaknesses:**

Strengths
- Model is trained purely in the latent space, which makes training faster and allows optimization across the full noise range.
- Additional novel components are employed such as VFE and MPCF which assist training and improve results.
- Extensive experimental results across multiple datasets, models (SD 1.5 and SDXL) and human evaluation.
- Detailed ablation studies and hyper-parameter turning, where results such as the full noise range having the best score makes a strong case in favor of LPO.

Weaknesses
- The method involves extensive hyperparameter tuning, such as threshold schedules and filtering strategies. This could limit robustness when transferred to different architectures or datasets.
- There are no specific failure cases or qualitative examples demonstrating where LPO underperforms when using the recommended hyperparameters. This limits insight into potential failure modes and areas for future improvement.
- MPCF is a data filtering strategy and while it is shown to be effective, it is influenced by the similar scores it is evaluated against (Aesthetic is also in eval). This can be a concern if there are similar biases between the two groups (filtering and eval metrics).

---

> ### Author Rebuttal · Authors · 2025-07-31
>
> Thanks for your effort in reviewing our paper and giving appreciation regarding our novelty, methods, and experiments. We hope the following responses address your concerns.
>
> > Q1 (Weakness 1): The method involves extensive hyperparameter tuning, such as threshold schedules and filtering strategies. This could limit robustness when transferred to different architectures or datasets.
>
> R1: Thanks for your valuable question. Your concern is very reasonable, as an excessive number of hyperparameters can indeed affect the robustness of a method. In this work, we conducted experiments to verify the generalization of the proposed method and hyperparameters across datasets and model architectures.
>
> - For **datasets**, we used exactly the same hyperparameters as those in the experiments on Pick-a-Pic, including MPCF, VFE, and Dynamic Threshold, and performed experiments on HPDv2 (Tables 15, 16, and 17 in the appendix), confirming the effectiveness of both the method and the hyperparameters.
> - For **model architectures**, we applied the same MPCF strategy (with the same hyperparameters) on SD1.5, SDXL, and SD3, and again verified the method's effectiveness. However, for different models, we used different Dynamic Threshold hyperparameters, as this parameter is influenced by the distribution of $x_t$ in each model. **We also complement extra hyperparameter experiments on SDXL to validate their generalization**, as detailed in **R6**.
>
> Overall, these experiments demonstrate that, with no adjustment or minor adjustment of a small subset of hyperparameters, our methods generalize well across datasets and model architectures.
>
>
>
> > Q2 (Weakness 2 & Question 1): No specific failure cases and their corresponding analysis.
>
> R2: Thanks for your constructive suggestion. We regret that the current paper lacks the visualization and analysis of failure cases. During our research, we observed the following types of failure cases:
>
> - **Reward hacking**: When training steps are excessive, the model exhibits reward hacking, where the reward metric continues to improve while the quality of generated images degrades. This stems from a certain misalignment between the reward model and human preferences, which remains a common issue across most existing methods. Mitigating this problem requires a combination of strategies, including improvements in preference data and reward modeling approaches.
> - **Decline in diversity**: Extended training also leads to a reduction in generation diversity. This is caused by the reward model narrowing the output distribution of the diffusion model. A general solution is prompt engineering, such as rephrasing input prompts to enhance the diversity of generated images via prompt variation [1].
> - **Lack of fine details on SDXL**: For some prompts, SDXL models optimized with LPO generate images with less detailed content compared to those optimized with SPO. This phenomenon was not observed on SD1.5. We will investigate this in future work to identify the underlying causes.
>
> Due to the rebuttal policy, we are unable to include images, external links, or upload new PDFs in our response. **We will include the above analysis and corresponding image examples in the revised version.**
>
>
>
> > Q3 (Weakness 3): MPCF is a data filtering strategy and while it is shown to be effective, it is influenced by the similar scores it is evaluated against (Aesthetic is also in eval). This can be a concern if there are similar biases between the two groups (filtering and eval metrics).
>
> R3: Thanks for your insightful question. This is a very reasonable concern. In our experiments, we also observed a certain correlation between the aesthetic scores used in filtering strategies and in evaluation metrics. For example, strictly enforcing aesthetic constraints (the first strategy in Table 1) leads to aesthetic hacking (Lines 206–207), and as a result, the aesthetic score in the final evaluation metric is also higher (the first strategy in Table 7).
>
> However, since we have not yet found a better open-source reward model for the aesthetic dimension, we still used the Aesthetic Predictor for automatic evaluation. **To provide a more objective assessment, we also conducted multi-dimensional (inlcuding aesthetic) human evaluations for the final model (Figure 6 in the appendix). The results of the user study are consistent with those from the automatic evaluation.** We will also consider using different aesthetic models for filtering and evaluation in future work.
>
>
>
> > Q4 (Question 2): Concerns with the speed comparison.
>
> R4: Thanks for your detailed question. In Table 6, we have listed the time for reward modeling and preference optimization separately, as well as their total time. For Diffusion-DPO, we set the reward modeling time to 0. In fact, in the paper, **the training time we refer to (Line 15, Line 74, Lines 290–292) always includes both reward modeling and preference optimization**, so the comparison is fair.
>
>
>
> > Q5 (Question 3): I was also not able to find the DPO training time figures so I think a specific source should be included or if the authors reproduced it themselves they should provide details for this.
>
> R5: Thanks for your question. The training time of Diffusion-DPO is provided in Table 6. We reproduced the results using the open-source repository (you can search DiffusionDPO on GitHub; we cannot post the direct URL due to the rebuttal policy) with 16 A100 GPUs, a local batch size of 1, and a gradient accumulation step of 128. Therefore, the effective batch size is 2048, consistent with that claimed in the original paper [2]. The number of training steps is 2000.
>
>
>
> > Q6 (Question 4): While the ablation studies are extensive and well executed, they are conducted only on SD1.5. However SDXL and SD1.5 can be very different. It will be beneficial to perform threshold, VFE and MPCF experiments on SDXL to show that hyper-parameters generalize across model scales.
>
> R6: Thanks for your valuable suggetion. We supplement ablation experiments on SDXL involving MPCF, VFE, and Dynamic Threshold. The results are provided as follows, **exhibiting similar patterns and trends as those observed on SD1.5 (Tables 7, 8, and 10)**. This further demonstrates the generalizability of our methods and hyperparameter settings across models of different scales and architectures.
>
> - MPCF
>
> |Strategy|Aes-Corr|CLIP-Corr|Aesthetic|GenEval|PickScore|
> |-|-|-|-|-|-|
> |wo MPCF|0.1355|0.1052|6.134|56.33|22.63|
> |1|**0.4528**|0.1716|**6.629**|57.56|22.72|
> |2|0.0729|0.4032|6.360|**59.27**|**22.86**|
> |3|-0.0698|**0.4656**|6.137|59.00|22.68|
>
> - VFE
>
> |VFE|gs|Aes-Corr|CLIP-Corr|Aesthetic|GenEval|PickScore|
> |-|-|-|-|-|-|-|
> |$ \times $|1.0|**0.1534**|0.3314|**6.381**|58.33|22.73|
> |$ \checkmark $|3.0|0.0989|0.3912|6.369|58.76|22.81|
> |$ \checkmark $|7.5|0.0729|0.4032|6.360|**59.27**|**22.86**|
> |$ \checkmark $|10.0|0.0632|**0.4147**|6.322|59.03|22.77|
>
> - Dynamic Threshold
>
> |Threshold|PickScore|ImageReward|HPSv2|HPSv2.1|Aesthetic|GenEval|
> |-|-|-|-|-|-|-|
> |0.5|22.77|1.1838|28.73|31.46|6.327|58.48|
> |0.6|22.79|1.1535|28.65|31.64|6.323|57.19|
> |[0.4, 0.55]|22.81|**1.2387** |28.86|31.52|6.354|58.72|
> |[0.45, 0.6]|**22.86**|1.2166|**28.96**|**31.89**|**6.360**|**59.27**|
> |[0.5, 0.65]|22.79|1.2091|28.95|31.74|6.277|58.68|
>
>
>
> > Q7 (Limitation 1): Should add "Since LPO is performed within the latent space, which is determined by the VAE encoder, the VAE of LRM should be identical to that of DMO."
>
> R7: Thanks for your valuable suggestion. We will include this statement in the "Limitation" section in the revised version.
>
>
>
> > Q8 (Limitation 2): The training process uses MPCF, which filters data based on scores like Aesthetic Score, CLIP Score, and VQAScore. This process can inherit any issues that are shared across these metrics.
>
> R8: Thanks for your insightful suggestion. As you pointed out, the data filtered by MPCF is influenced by the filtering metrics used (Aesthetic, CLIP Score, and VQA Score). The combined use of multiple metrics helps reduce this risk, but it still persists. We will include a discussion of this limitation in the revised version.
>
>
>
> > Q9: Typos
>
> R9: Thanks very much for your detailed review. We will correct the following typos in the revised version.
>
> - Table 23: "Ablataion" -> "Ablation"
> - Line 150: "averagely" -> "average pooled"
>
>
>
> [1] Gong L, Hou X, Li F, et al. Seedream 2.0: A native chinese-english bilingual image generation foundation model. arXiv preprint arXiv:2503.07703, 2025.
>
> [2] Wallace B, Dang M, Rafailov R, et al. Diffusion model alignment using direct preference optimization. CVPR, 2024.

---

> ### Comment · Reviewer_Xmnx · 2025-08-04
>
> Thank you for the additional clarifications and experiments. I am willing to keep my score.

---

> > ### Author Response · Authors · 2025-08-04
> >
> > Dear Reviewer Xmnx,
> >
> > Thank you for your time and thoughtful feedback. We sincerely appreciate your positive assessment and your willingness to maintain your score. Your support and suggestions are valuable to us, and we are committed to further improving our work based on all reviewers’ comments.

---

### Official Review · Reviewer_aRG8 · 2025-07-02

**Clarity:** 3
**Significance:** 2
**Originality:** 2
**Rating:** 4
**Confidence:** 3

**Summary:**

The authors proposed LPO, an novel algorithm that leverages latent diffusion backbone as a powerful and efficient reward model that learns to rank noisy images in the latent space. Sampling inference strategies including multi-preference filtering, dynamic thresholding has been introduced for efficient online sampling. LPO achieved superior results comparing to multiple baselines in both general image quality and human preference alignment on SD1.5, SDXL. Further studies also demonstrated that LPO can be generalized to flow matching objective and DiT structure.

**Questions:**

* Do the authors decide MPCF strategies based on heuristics? Do you think it's possible to develop a principled learned MPCF filtering pipeline?

* Pick a Pic have 5M image pairs, may I know how long does it take to filter the image pairs?

* Table 19 and Table 20 seems a bit unfair comparison as pick score has a smaller size with CLIP + small MLP, while even smallest SD1.5 LRM is CLIP + Unet. It's hard to tell how much latent space reward modeling itself improves alignment performance.

**Ethical Concerns:**

["NO or VERY MINOR ethics concerns only"]

**Final Justification:**

See my comments

**Limitations:**

yes

**Paper Formatting Concerns:**

No major concerns

**Quality:**

3

**Strengths And Weaknesses:**

* Strengths:
  * The authors conduct comprehensive experiments across multiple evaluation metrics of T2I image synthesis and verify the proposed method with base T2I models covering various architectures and training objectives
 * LPO demonstrated robust and superior results against baselines

* Weaknesses:
  * The novelty of the proposed method is limited, the proposed method essentially leverages the SD-VAE to allow the reward model training to latent space and employed stronger filtering mechanism to further improve alignment performance, the technical contribution is limited.
 * Poor generalizability: Even though LRM 1.5 was proved effective on SD2.1, the success can be attribute more to the similar capability of the two models as LRM 1.5 was ineffective for SDXL. Therefore, the proposed method needs to fine-tune diffusion backbone as reward model before alignment if the base model differs significantly from a pre-trained LRM, adding complexity in potential tuning to the post-training pipeline.

---

> ### Author Rebuttal · Authors · 2025-07-31
>
> Thanks for your effort in reviewing our paper and giving appreciation regarding our methods and experiments. We hope the following responses solve your concerns.
>
> > Q1 (Weakness 1): The novelty of the proposed method is limited, the proposed method essentially leverages the SD-VAE to allow the reward model training to latent space and employed stronger filtering mechanism to further improve alignment performance, the technical contribution is limited.
>
> R1: Thanks for your question. There may be some misunderstanding of the novelty and contributions of our work. We respond from the following aspects:
>
> - First, the proposed LRM does more than shift reward model training from the pixel space to the clean latent space via the VAE. More importantly, **LRM leverages the noise-aware ability of pre-trained diffusion models to shift the reward model training from the clean latent space to the noisy latent space**. This enables LRM to understand noisy latent images $x_t$, resulting in significantly better noise compatibility (as shown in Table 21), and faster reward model training and preference optimization (Tables 5 and 6).
>
> - Second, while LLM-as-a-judge [1] has been widely adopted in text generation, in image generation, our work is the **first** to use the diffusion model itself for reward modeling and preference optimization. **It demonstrates superior effectiveness and efficiency compared to VLM-based methods, as appreciated by Reviewer Xmnx, rhE9, and 2Nf4**. Just as diffusion models have opened new directions in language modeling [2], our work pioneers a novel path in reward modeling for visual generation. We believe this area holds great potential and deserves further exploration.
>
> - Third, although MPCF is a filtering strategy, it reflects our in-depth thinking on step-level reward modeling. SPO [3] assumes that preferences vary across timesteps. However, the reward model training of SPO assumes preferences remain unchanged when noise is added to image pairs at various timesteps, creating an **evident conflict**. Based on this observation, we propose the MPCF strategy to filter the data and mitigate the conflict between the assumptions of step-wise preference optimization and reward model training.
>
> In summary, we sincerely hope you may reconsider the novelty and contributions of this work to the research community.
>
>
>
> > Q2 (Weakness 2): Poor generalizability: Even though LRM 1.5 was proved effective on SD2.1, the success can be attribute more to the similar capability of the two models as LRM 1.5 was ineffective for SDXL. Therefore, the proposed method needs to fine-tune diffusion backbone as reward model before alignment if the base model differs significantly from a pre-trained LRM, adding complexity in potential tuning to the post-training pipeline.
>
> R2: Thanks for your question. As stated in Lines 238–239, the VAE of LRM must be identical to that of the diffusion model to be optimized. Therefore, LRM-1.5 is effective on SD2.1 (same VAE), but ineffective on SDXL (different VAE). Regarding the need to fine-tune a reward model for each diffusion model, we respond from the following two perspectives:
>
> - First, for step-level reward models, fine-tuning is necessary when the base diffusion models differ significantly. Even at the same timestep *t*, the distribution of $x_t / \hat{x}\_{0,t} / I_t$ for the same prompt differs across models. Therefore, the reward model must be aligned with the $x_t / \hat{x}\_{0,t} / I_t$ distribution of each diffusion model. **For example, SPO [3] also trains separate SPMs for SD1.5 and SDXL**.
>
> - Second, even for reward models that only take $I_0$ as input, the image distributions generated by different diffusion models can vary considerably. To our knowledge, in current industrial practice [4,5], **it is common to separately fine-tune reward models using images sampled from the current diffusion model to be optimized**. This helps adapt the reward model to the image distribution of the current diffusion model and enables mitigation of model-specific failure cases. It is uncommon to use the same reward model to optimize significantly different diffusion models simultaneously.
>
> Considering the above, **we believe that LRM does not introduce significant additional training complexity compared to other reward models.** In addition, extensive experiments in the paper demonstrate that **our method generalizes well across different model architectures (SD1.5, SDXL, SD3), datasets (Pick-a-Pic, HPDv2), and optimization methods (LPO, GRPO).**
>
>
>
> > Q3 (Question 1): Do the authors decide MPCF strategies based on heuristics? Do you think it's possible to develop a principled learned MPCF filtering pipeline?
>
> R3: Thanks for your valuable question. MPCF employs three pre-trained public reward models to filter data, with the threshold for the score gap $ G_* $ chosen according to experimental results. In this work, we select $ G_* $ by evaluating the performance of LRM trained on the filtered data. Our goal is to encourage LRM to attend to both image aesthetics and image-text alignment, as indicated by Aes-Corr, CLIP-Corr, and the performance of models optimized via LRM and LPO. We would greatly appreciate it if you could provide more details about "a principled learned MPCF filtering pipeline," and we are willing to follow up on your further suggestions.
>
>
>
> > Q4 (Question 2): Pick a Pic have 5M image pairs, may I know how long does it take to filter the image pairs?
>
> R4: Thanks for your question. In fact, the Pick-a-Pic v1 dataset [6] contains 0.58M image pairs, not 5M, as stated in Lines 204–205. For MPCF filtering, we used 8 A100 GPUs: the Aesthetic Predictor took 0.7 hours, CLIPScore 0.9 hours, and VQAScore 5 hours, resulting in a total of 52.8  ($(0.7+0.9+5)\times8$) A100 GPU hours. Notably, this is a one-time cost, and the filtered data can be reused across different model architectures and hyperparameter settings, including SD1.5, SDXL, and SD3. **Considering the speed comparison in Table 6, even after including this one-time computational cost, the overall training time of our method remains lower than that of Diffusion-DPO and SPO**.
>
>
>
> > Q5 (Question 3): Table 19 and Table 20 seems a bit unfair comparison as pick score has a smaller size with CLIP + small MLP, while even smallest SD1.5 LRM is CLIP + Unet. It's hard to tell how much latent space reward modeling itself improves alignment performance.
>
> R5: Thanks for your question. **In fact, the parameter count of LRM-1.5 is less than half that of PickScore**. We respond from the following two perspectives.
>
> - PickScore [6] and SPM in SPO [3] are both finetuned from CLIP-H, whose vision and text encoders together contain over **980M** parameters. In contrast, LRM-1.5 uses the CLIP-L text encoder (123M parameters) and only the down and mid blocks of the U-Net (346M parameters), resulting in a total of **~470M** parameters, **which is less than half the size of PickScore**.
>
> |Model|Number of Parameters|
> |-|-|
> |PickScore (SPM)|630M (CLIP-H Vision Encoder) + 353M (CLIP-H Text Encoder) + Small MLP|
> |LRM-1.5|123M (CLIP-L Text Encoder) + 346M (U-Net Down & Mid Block) + Linear Projection Layer|
>
> - We also conduct experiments on two internal DiT‑based models. We compare two different reward models: one is LRM, which utilizes parts of the parameters of the DiT, and the other is a VLM with more than 25B parameters. They are trained on internal image‑text alignment datasets and then employed for the preference optimization of the DiT models. For the same diffusion model, we train two reward models using identical data. The human‑evaluation results for image‑text alignment of optimized DiT models are as follows:
>
> | Diffusion Model | Reward Model | Alignment |
> | --------------- | ------------ | --------- |
> | DiT-S      | LRM (9.5B)   | **72.5**  |
> | DiT-S      | VLM (>25B)   | 71.2      |
> | DiT-L         | LRM (15B)    | **82.8**  |
> | DiT-L        | VLM (>25B)   | 80.8      |
>
> These results consistently demonstrate that reward modeling in the noisy latent space, as done by LRM, can significantly enhance alignment performance **while being more parameter-efficient**.
>
>
>
> [1] Gu J, Jiang X, Shi Z, et al. A survey on llm-as-a-judge. arXiv preprint arXiv:2411.15594, 2024.
>
> [2] Nie S, Zhu F, You Z, et al. Large language diffusion models. arXiv preprint arXiv:2502.09992, 2025.
>
> [3] Liang Z, Yuan Y, Gu S, et al. Aesthetic post-training diffusion models from generic preferences with step-by-step preference optimization. CVPR 2025.
>
> [4] Gong L, Hou X, Li F, et al. Seedream 2.0: A native chinese-english bilingual image generation foundation model. arXiv preprint arXiv:2503.07703, 2025.
>
> [5] Gao Y, Gong L, Guo Q, et al. Seedream 3.0 technical report. arXiv preprint arXiv:2504.11346, 2025.
>
> [6] Kirstain Y, Polyak A, Singer U, et al. Pick-a-pic: An open dataset of user preferences for text-to-image generation. NeurIPS, 2023.

---

> > ### Author Response · Authors · 2025-08-04
> >
> > Dear Reviewer aRG8,
> >
> > Thank you again for your time and effort in reviewing our paper. As the discussion phase is drawing to a close, we would like to kindly ask whether our rebuttal has adequately addressed your concerns. If there are any remaining questions or points that need further clarification, we would be happy to provide additional details within the remaining time. If you feel that your concerns have been resolved, we would sincerely appreciate it if you would consider updating your score to reflect the newly added results and discussion. We would also greatly value the opportunity to continue the discussion if needed.

---

> > > ### Comment · Reviewer_aRG8 · 2025-08-04
> > >
> > > I appreciate the author's efforts in the rebuttal. My concerns are revolved and I will increase the rating accordingly.

---

> > > > ### Author Response · Authors · 2025-08-05
> > > >
> > > > Dear Reviewer aRG8,
> > > >
> > > > Thank you very much for your thoughtful feedback and for taking the time to review our rebuttal. We sincerely appreciate your support for our paper and are grateful that our response effectively addressed your concerns.
> > > >
> > > > We noticed that, although you mentioned increasing the rating, the updated score does not yet seem to be reflected in the system. If this is intentional, please feel free to disregard this note. We just want to gently check in case it is missed unintentionally. Once again, we truly appreciate your valuable comments and support.
> > > >
> > > > Best regards!

---

### Note · Authors · 2025-08-11

Dear Reviewers and Area Chair,

During the rebuttal phase, we diligently addressed the concerns raised by all four reviewers. **After a thorough round of discussions, each reviewer kindly acknowledged that their concerns had been effectively resolved. As a result, their scores were either increased or remained positively oriented.** We sincerely appreciate the reviewers' insightful feedback and the Area Chair's guidance, which have significantly contributed to the improvement of our work.

Thank you once again for your time and valuable input.

---

### Decision · Program_Chairs · 2025-09-17

**Decision:**

Accept (poster)

**Comment:**

This paper introduces a reward model for Diffusion preference optimization that is defined on the latent space of the diffusion model itself, and uses elements of it, as opposed to the traditional use of VLMs. They additionally refine the preference learning algorithm with ideas for preference pair filtering. The resulting algorithm leads to improvements in visual quality and alignment, along with training time speedups.

Strengths: Use of the latent space  to predict rewards in this way is novel. The key insight that preferences over final images may not be consistent with preferences of the noised images at each time-step is important. Reviewers appreciated the contributions in the algorithm design as well. Overall, small but consistent gains were achieved.

There were no serious weaknesses raised by reviewers that were not satisfied by the end of the discussion. One downside of the method is that the RM has to be re-trained to align with each generative model separately.

Overall this seems like a refinement of existing approaches that is well justified and useful for practitioners in this popular field.